# Assessing the Seasonal Evolution of Snow Depth Spatial Variability and Scaling in Complex Mountain Terrain

Zachary S. Miller[1], Erich H. Peitzsch[1], Eric A. Sproles[2], Karl W. Birkeland[3], and Ross T. Palomaki[2]

[1]US Geological Survey Northern Rocky Mountain Science Center, West Glacier, Montana, MT 59936, USA

[2]Geographic Snow, Water, and Ice Resources Lab, Department of Earth Sciences, Montana State University, Bozeman, Montana, MT 59717, USA

[3]USDA Forest Service National Avalanche Center, Bozeman, Montana, MT 59771, USA

*Correspondence to*: Zachary S. Miller (zsmiller@usgs.gov)

**Abstract.** Dynamic natural processes govern snow distribution in mountainous environments throughout the world. Interactions of these different processes create spatially variable patterns of snow depth across a landscape. Variations in accumulation and redistribution occur at a variety of spatial scales, which are well established for moderate mountain terrain. However, spatial patterns of snow depth variability in steep, complex mountain terrain have not been fully explored due to insufficient spatial resolutions of snow depth measurement. Recent advances in uncrewed aerial systems (UAS) and structure from motion (SfM) photogrammetry provide an opportunity to map spatially continuous snow depths at high resolution in these environments. Using UAS and SfM photogrammetry, we produced 11 snow depth maps at a steep couloir site in the Bridger Range of Montana, USA, during the 2019-2020 winter. We quantified the spatial scales of snow depth variability in this complex mountain terrain at a variety of resolutions over two orders of magnitude (0.02 m to 20 m) and time steps (4 to 58 days) using variogram analysis in a high-performance computing environment. We found that spatial resolutions greater than 0.5 m do not capture the complete patterns of snow depth spatial variability within complex mountain terrain and that snow depths are autocorrelated within horizontal distances of 15 m at our study site. The results of this research have the potential to reduce uncertainty currently associated with snowpack and snow water resource analysis by documenting and quantifying snow depth variability and snowpack evolution on relatively inaccessible slopes in complex terrain at high spatial and temporal resolutions.

## 1 Introduction

Seasonal mountain snowfall is a critical natural resource globally but can also present a natural hazard for mountainous communities. Understanding the spatial distribution and temporal evolution of seasonal snow depth, defined as the vertical distance from the snow surface to the base of the snowpack (Fierz et al., 2009), is essential for water resource managers, local governments, climate researchers, and avalanche forecasters. However, quantifying snow depth across a landscape, especially one comprised of mountainous terrain, is challenging due to the multi-scalar nature of physical processes governing the distribution of snow depth (Blöschl, 1999; Bühler et al., 2016; Egli et al., 2011; Elder et al., 1998; Grünewald et al., 2010;

Liston et al., 2007; Schweizer et al., 2008; Trujillo et al., 2009). These physical processes interact in different ways throughout the landscape, influencing the local spatial variability of snow depth in non-homogenous ways over spatial scales ranging from less than a centimeter to 100 meters (m) or greater. The international snow depth monitoring community follows guidelines for selecting research sites in wind-sheltered, flat locations (Buchmann et al., 2021). Although such relatively homogenous terrain allows for clearer differentiation of some specific processes influencing snow depth distribution, such as wind-vegetation interactions (Deems et al., 2006; Trujillo et al., 2007; Trujillo et al., 2009), previous research overlooks steep, complex slopes, an essential characteristic of mountainous terrain, where much of the seasonal snowpack exists (Deschamps-Berger et al., 2020). In this study, we define the slope scale as a spatial extent of $< 0.2 \text{ km}^2$ and complex terrain as mountainous topographies including hillslopes $> 25°$ with interspersed rock outcrops, vertical cliff features, and variable slope geometries.

A major challenge in accurately analyzing the spatial variability of snow depth is acquiring measurements at an appropriate spatial resolution (Clark et al., 2011; Kinar and Pomeroy, 2015). Current methods for mapping snow depth in a spatially continuous manner within steeper mountain topographies are limited (López-Moreno et al., 2015; Meyer and Skiles, 2019). Traditional methods of measuring snow depth include in-situ snow surveys, snow pits, and automated weather stations (AWS), which provide a spatially incomplete measure of snow depth made up of sparse point measurements distributed heterogeneously over the landscape (Dozier, 2011; Dozier et al., 2016; Elder et al., 1998; Grünewald et al., 2010; López-Moreno et al., 2011). Point measurement locations typically avoid exposure to snow avalanches due to safety and logistical concerns, and therefore measurements collected from relatively flat, planar terrain are overrepresented compared to measurements from steeper slopes. Remotely sensed measurements, on the other hand, can acquire spatially continuous snow depth measurements across a variety of terrain at multiple resolutions without exposing observers to avalanches. Current satellite-derived snow depth data (e.g., Pléiades, WorldView-3, and WorldView-4) are easily accessed, spatially continuous, and, through stereo imagery processing, map snow depth at 2 m horizontal resolution with 0.5 m vertical accuracy (Deschamps-Berger et al., 2020; Hu et al., 2016; Marti et al., 2016). Yet the accuracy of DEMs produced through satellite imagery and stereoscopic processing are known to suffer on slopes steeper than 35°, common in high-relief mountain terrain (Lacroix, 2016; Shean et al., 2016, Deschamps-Berger et al., 2020). Therefore satellite imagery derived DEMs are still insufficient for capturing some of the finer-scale processes which influence snow depth distributions in complex terrain (Eker et al., 2019). Terrestrial laser-scanning can acquire spatially continuous centimeter-scale resolution snow depth data, but is limited by its inherent field of view, shadowing by steep topographic features (Deems et al., 2013; Fey et al., 2019; Prokop et al., 2015; Trujillo et al., 2007) and can be overly cumbersome for surveys in remote areas. Airborne laser-scanning (e.g., Airborne Snow Observatory - Painter et al., 2016) circumvents the terrain-shadowing shortcomings of terrestrial laser-scanning yet remains cost-prohibitive for many researchers (Brandt et al., 2020; Bühler et al., 2015; Dozier et al., 2016; Meyer and Skiles, 2019).

Imagery captured from uncrewed aerial systems (UAS) combined with structure-from-motion (SfM) photogrammetry techniques allows for low-cost collection of spatially continuous centimeter-scale resolution snow depth data with few terrain

limitations, making it an attractive tool for snow depth distribution mapping in complex, non-forested mountain terrain (Avanzi et al., 2018; Bühler et al., 2016; De Michele et al., 2016; Eberhard et al., 2021; Gaffey and Bhardwaj, 2020; Redpath et al., 2018; Revuelto et al., 2021). Numerous studies conclude that UAS and SfM techniques are effective at mapping snow depth variability at the slope scale, yet most focus on simpler terrain and only compare two individual timestamps of data (Adams et al., 2018; Avanzi et al., 2018; Boesch et al., 2016; Bühler et al., 2016; Cimoli et al., 2017; De Michele et al., 2016; Eberhard

et al., 2021; Gabrlik et al., 2019; Harder et al., 2016; McCormack and Vaa, 2019; Redpath et al., 2018; Peitzsch et al., 2018; Vander Jagt et al., 2015).

Seasonal snowpack is constantly evolving and the process scales at which it changes are variable throughout both space and time (Blöschl, 1999). The spatial and/or temporal resolutions of measurements in previous snow depth research have been

insufficient to capture the process scales of spatial heterogeneity in the evolving snowpack (Clark et al., 2011; López-Moreno et al., 2011). Here, we utilize 0.02 m horizontal resolution UAS/SfM derived snow depth observations as our baseline for further snow depth spatial variability analysis. The observation scales in this study span multiple orders of magnitude spatially (0.02 m grid covering approximately 0.2 km$^2$ extent) on 11 distinct observation days over five months. These scales allow us to observe centimeter-scale vertical changes in snow depth across the slope scale, here defined as $< 0.2$ km$^2$. Prior studies

demonstrate the value of the slope scale for exploring the complex nature of snow depth variability and understanding avalanche formation processes (Anderton et al., 2004; Birkeland, 2001; Birkeland et al., 1995; Kronholm and Birkeland, 2007; López-Moreno et al., 2015; Schweizer et al., 2008; Wirz et al., 2011). The temporal resolution of this study allows us to observe the evolution of snow depth spatial variability throughout the winter, in comparison to previous research that inferred patterns of snow depth spatial variability from more sparse temporal observations (López-Moreno et al., 2015; Niedzielski et al., 2019;

Mendoza et al., 2020).

Our study considers the question: What is the optimal sample spacing that fully captures snow depth variability at the slope scale in complex mountain terrain? The objective of this work is to quantify the optimal spatial resolution necessary for accurate representation of snow depth spatial variability and its seasonal evolution in the complex terrain of our study site. To

achieve this, we analyze the differences in patterns of snow depth between complex and relatively simple mountain terrain at the slope scale. We also investigate the temporal evolution of snow depth spatial variability throughout the course of a winter at our study site.

## 2 Research Site

The research site is a steep sub-alpine mountain basin within the Bridger Range of southwest Montana, USA (45.834° N, -

110.935° E), at the head of the South Fork Brackett Creek watershed (Fig. 1). The Bridger Range is classified as an intermountain snow and avalanche climate characterized by average December through March temperatures from -3.5 to -7°C

and average annual snowfall of approximately 7.5 m measured at Bridger Bowl Ski Area (Mock and Birkeland, 2000). The surrounding area (sometimes referred to as "wolverine basin"), has been the site of frequent snow and avalanche research over the past 20 years (Deems, 2002; Landry et al., 2004; Lundy et al., 2001; Van Peursem et al., 2016) due to its safe access, heterogeneous terrain, and proximity to Bridger Bowl Ski Area's network of automated weather stations (AWS).

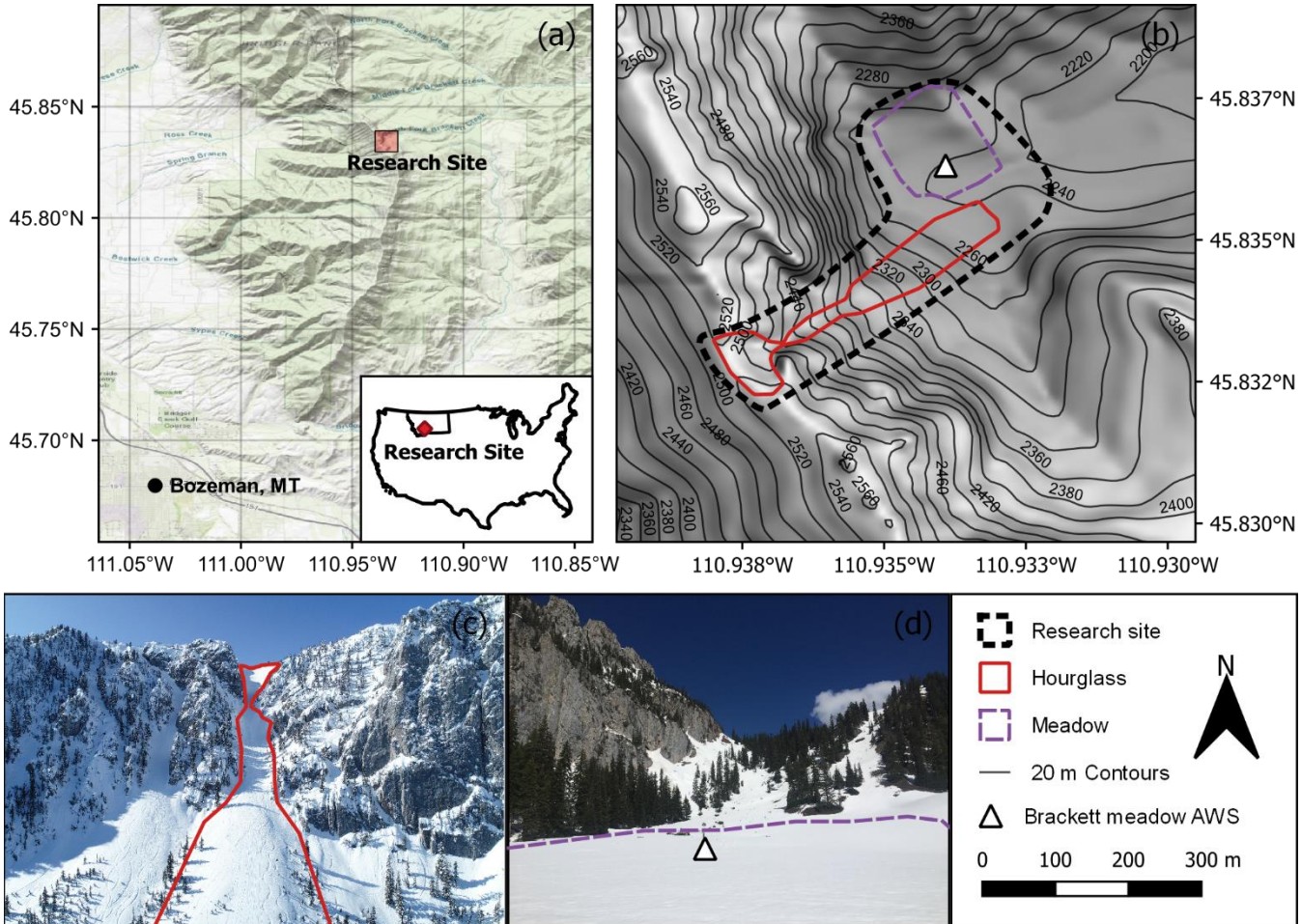

Figure 1: Study location in the Bridger Range of Montana, USA (a) and overview topographic map of the study area (b). The general research site (dotted polygon), the Hourglass couloir (solid red), the meadow (dashed purple), and the Brackett meadow AWS (white triangle) are shown. Photographs of the Hourglass couloir (c) and meadow (d) with the couloir and meadow polygons outlined. Data map source: U.S. Geological Survey, 2017.

There are two distinct mountain topographies, a steep couloir and sheltered meadow, within the research site that are subject to similar meteorological conditions. The Hourglass (2250 – 2550 m a.s.l.) is a couloir and avalanche path that has a mean slope angle of 33° and is mainly composed of rock scree, outcrops of limestone cliffs, and 1 - 2 m subalpine fir and Engelmann spruce trees. Mature, 10 - 20 m in height, coniferous trees and 1 m tall shrubs/bushes border the main avalanche path. Ridgetop

wind-loading from dominant westerly storms result in large cornice growth during the winter along the top of the couloir and frequent small natural avalanches within the avalanche path (Lundy et al., 2001). The Hourglass is infrequently skied due to its avalanche-prone terrain, and we observed approximately 5 unique ski tracks throughout our field season. The meadow (2240 m a.s.l.) is adjacent to the runout of the Hourglass, has a mean slope of 5°, and consists of a mix of grasses and shrubs with 10 - 20 m tall mature coniferous forests on its north, west, and south sides. The meadow is sheltered from all but easterly wind directions and localized severe weather by the surrounding dense forest and steep 300 m headwall to its west.

We used meteorological data from an AWS in the immediate vicinity of the research site for measuring snow depth and other related meteorological variables. The Brackett meadow AWS (2240 m), located in the meadow, measured hourly temperature, relative humidity, wind speed, wind direction, net radiation, and snow depth from 6 November 2019, to 10 June 2020. The Brackett meadow AWS's below-treeline, wind-protected location is similar to many SNOTEL sites (Molotch and Bales, 2006).

## 3 Methods

Our study aims to quantify optimal sample spacing to fully capture the spatial variability of snow accumulation and redistribution at the slope scale in complex mountain terrain. To achieve this goal, we first generated digital surface models (DSMs) with UAS-based SfM photogrammetry techniques collected on 11 field days during the 2019 - 2020 winter. We collected in-situ snow depth measurements via manual probe for validation. Then, we used the high resolution (0.02 m horizontal) DSMs and resampled at coarser resolutions to calculate multi-resolution variograms to assess the scales of spatial variability patterns in snow depth. Finally, we used scene-wide coefficient of variation calculations to analyze seasonal patterns in snow depth spatial variability.

### 3.1 UAS Surveys

We designed our aerial surveys to achieve horizontal spatial resolutions of < 0.05 m to observe centimeter-scale differences in snow depths throughout the entire study site (De Michele et al., 2016; Fierz et al., 2009). We used a commercially available DJI Phantom 4 RTK UAS equipped with a 20-megapixel camera and Real-Time-Kinematic (RTK) Global Navigation Satellite System (GNSS) with the DJI DRTK2 GNSS mobile station. We conducted repeated autonomous pre-programmed UAS missions flying in a grid pattern at a constant 50 m above ground level based on a 1 m resolution DSM collected prior to winter flights. Our flight imagery was collected with 70% front/side image overlap, resulting in < 2 cm per pixel average ground sampling distance and approximately 550 overlapping images each field day, similar to Goetz & Brenning (2019). A data gap exists between 17 March 2020 and 14 May 2020, due to the onset of the COVID-19 pandemic.

To constrain topographic error in post-processing, we collected 25 stationary and easily recognizable ground control points with high-resolution RTK GNSS survey equipment during the snow-free season to incorporate into the digital surface models.

Due to variable snow cover, we used a partial selection of the 25 ground control points, typically 3 - 10 points, in each individual model by selecting points that were not covered by snow. To constrain snow depth observation error, each field day we deployed and surveyed at least 4 snow depth validation point targets prior to flights in safe accessible locations within the study area and manually measured snow depths at each point immediately after the flights (described further in Sect. 3.3).

## 3.2 Digital Surface Model (DSM) Creation

For each field date, we processed the overlapping imagery to derive snow depth maps using three steps: Post-Processing-Kinematic (PPK) location corrections, SfM processing of imagery for DSM creation, and DSM-differencing to derive snow depth.

We post-processed the UAS location data to improve the quality of the RTK GNSS positions and ensure accurate co-registration of output models using RTKLIB (Takasu, 2009) and the R software environment (R Core Team, 2021) in the WGS84 geographic coordinate system (EPSG::4326). Using the MTSU reference station (~21 km from the study site), the National Aeronautics and Space Administration's Daily Global Positioning Systems broadcast ephemeris data, and the UAS RINEX and Timestamp files, our PPK processing resulted in horizontal positional accuracies of less than 0.1 m and vertical positional accuracies of less than 0.2 m for most UAS photo locations (Table A1). This additional step was necessary due to limited satellite connectivity of the RTK system from the poor sky view of high-relief topography at our study site.

We completed SfM photogrammetric processing using the software package Agisoft Metashape Pro Version 1.6.2/1.6.6 (Agisoft, 2020), which generates 3-D surface models from overlapping imagery and point matching (Alidoost and Arefi, 2017; Carbonneau and Dietrich, 2017; Gabrlik et al., 2018; Nolan et al., 2015). We filtered, aligned, and reduced the error of the geolocated imagery before the addition of ground control points for final batch processing. Finally, we ensured accurate co-registration by aligning the vertical and horizontal positions of the snow-covered models to available snow-free ground control points and the snow-free 8 July 2020 model (Adams et al., 2018). Utilizing fewer ground control points and UAS equipped with RTK provides similar accuracies as traditional ground control point driven SfM workflows (Eberhard et al., 2021; Revuelto et al., 2021). This processing workflow produced a DSM interpolated from a dense point cloud and an orthomosaic for further analysis. We used simple DSM-differencing techniques to calculate snow depths throughout our site by subtracting a snow-free DSM collected on 8 July 2020, from each snow-covered DSM, resulting in snow depth DSMs used for further spatial variability analysis. We removed poor quality DSMs if deemed unacceptable through comparison with probed snow depths, visual inspection, and expert judgment (Table A2). Examples of these thresholds include observed limited point matching while processing, inaccurate DSM reconstruction surfaces, unrealistic snow depths and unrealistic snow depth distributions. Unrealistic snow depths are negative snow depth values and values filtered by expert judgement.

### 3.3 Manual Snow Depth Collection

We collected traditional manual snow depth measurements through in-situ probing primarily within the lower elevations of the research area. These geolocated validation point snow depths were used for error assessment of UAS-derived snow depths We collected manual snow depth measurements at 4 or more random locations within avalanche-safe areas of the study site at deployed 1 m$^2$ markers each field day. To determine the 1 m$^2$ average, variation, and range at validation points, we probed manual in-situ snow depths at the center and four corners of each deployed marker (López-Moreno et al., 2011). We used the 1 m$^2$ averages as our individual validation point measurements. We collected location information for each of these manual snow depth measurements with handheld GPS units (3 - 5 m accuracy). Additionally, we manually collected a single in-situ avalanche crown profile in the upper elevations of the couloir on 28 February 2020, which included total snow depth, 25 individual snow layer thicknesses, grain type and size measurements, and an extended column test (ECT) as per Greene et al. (2016). We calculated summary statistics for manual snow depth measurements and their UAS-derived equivalent depths for each field day (Table A3).

We completed an assessment of error by calculating mean, standard deviation, root mean square error (RMSE) (Eq. 1) and normalized median absolute deviation (NMAD) (Eq. 2) values for the differences between UAS-derived and probed snow depths for both the complete set of snow depth DSMs and a subset with poor quality models of the meadow removed. RMSE is defined as

$$RMSE = \sqrt{\frac{1}{n}\sum_{i=1}^{n}(P_i - O_i)^2} \tag{1}$$

where *n* is the number of observations, *P* is the predicted value and *O* is the observed value, and NMAD is represented by

$$NMAD = 1.4826 \times \text{median}(|x_i - x_{median}|) \tag{2}$$

where 1.4826 is the scale factor for comparison with standard deviation, $x_i$ is the difference in measured snow depths for point *i,* and $x_{median}$ is the median of the dataset of differences.

Our error assessment followed a condensed version of the accuracy and precision measures presented in Adams et al. (2018) and Eberhard et al. (2021). The mean difference and RMSE are common measures of accuracy. Standard deviation and NMAD are common measures of precision, with NMAD being more resistant to outliers. We extracted the 1 m$^2$ median and inner quartile range (IQR) values from the corresponding snow depth DSM for each manually collected snow depth measurement location and used the median value for error assessment.

### 3.4 Digital Surface Model (DSM) Detrending

In preparation for variability analysis, we detrended each snow depth DSM with regular grids to focus analysis on the resultant residual surfaces as per Lutz and Birkeland (2011). Detrending allowed us to calculate omnidirectional variograms. First, we

masked individual vegetation features, such as trees, out of all DSMs while attempting to retain some snow surface between features. We used QGIS version 3.20.1 (QGIS.org, 2021) and the SAGA-GIS plugin's DTM Filter tool (Vosselman, 2000) to

205 filter out localized vertical spikes in elevation in the snow-free 8 July 2020, DSM and apply a 0.1 m buffer along the masked feature boundaries to account for minor vegetation shifts due to wind or snow creep (Table A3). We checked this mask against high resolution orthoimagery produced in the SfM workflow to ensure accuracy and applied this mask to each DSM included in our analysis.

We created detrended surfaces for each DSM using elevation, aspect, and distance-from-ridge as independent variables potentially contributing to snow depth trends. We extracted elevation, aspect, and distance-from-ridge raw surfaces from the snow-free 8 July 2020, DSM. Then, we calculated trend surfaces by selecting the most significant (lowest p-value) independent variable resulting from a least-squares linear regression for snow depth and each independent variable for each individual DSM. We subtracted the resultant trend surface from the raw snow depth DSM to produce detrended residual DSMs for each

field day. If no independent variables proved significant ($p > 0.05$), we detrended the DSMs by subtracting the mean snow depth from the raw snow depth DSM instead. A final correction using a 3 standard deviation filter and expert judgment removed erroneous outlier data from the detrended residual snow depth DSMs (Höhle and Höhle, 2009).

**3.5 Variogram Calculation and Fit**

To examine the spatial relationships of snow depth distributions in our two study sites, we used variogram analysis. Variograms

are useful for determining spatial structure and correlation of variables whose scaling behavior is unknown. Variograms are a visual representation of semivariance values calculated between point pairs at a variety of lag distances. The experimental variogram can be calculated as

$$\hat{\gamma}(h) = \frac{1}{2N(h)} \sum_{(i,j)}^{N(h)} (z_j - z_i)^2 \tag{3}$$

where $N(h)$ is the number of point pairs at the given lag distance $h$, and $z_i$ and $z_j$ are detrended snow depth values from individual

points separated by a lag distance $h$ (Webster and Oliver, 2007). The resultant semivariance values $\hat{\gamma}$ can be plotted against their lag distances and we can determine the separation distance $h$ at which point pair values are still correlated. This autocorrelation point is defined as the "sill" in terms of semivariance $\hat{\gamma}$ and the "range" in terms of lag distance $h$. The sill represents the overall variance of the input data. The range represents the maximum distance where point values are still correlated and is generally calculated as the lag distance at 95% of the sill. Point pairs further apart than the range are considered

not correlated and spatially independent. The nugget represents the potential measurement error and is the variance resulting from measurement error and natural variation found over distances shorter than the minimum sampling resolution. We calculated experimental omnidirectional variograms of the detrended residual snow depth DSMs at a variety of spatial resolutions and fit spherical models in the R software environment (R Core Team, 2021). Spherical models are well suited for

three-dimensional spatial analysis, are commonly used in similar variogram analyses, and fit the majority of our 170 experimental variograms (Kronholm, 2004; Kronholm et al., 2004; Kronholm and Birkeland, 2007; Webster and Oliver, 2007). Additionally, alternative models, such as exponential, Gaussian, and log-log linear, were utilized to fit snow depth spatial variability variograms in previous studies (Mendoza et al., 2020).

First, we resampled the detrended residual DSMs from their original 0.02 m spatial resolution to 0.05, 0.1, 0.25, 0.5, 1, 2.5, 5, 10, and 20 m horizontal spatial resolutions using four resampling methods: nearest-neighbor (Schön et al.; 2015), cubic convolution, mean aggregation, and median aggregation. We used paired-point correlations of the nearest neighbor resampled results with aggregated mean, aggregated median, and cubic convolution resampling techniques to compare the effects of each resampling method on variability calculations. We chose the nearest neighbor resampling technique to avoid over-smoothing observed using an aggregation or a cubic convolution resampling technique, and to avoid the uncertainty associated with the possibility of out-of-range values calculated through cubic convolution techniques (Roy and Dikshit, 1994; Fassnacht and Deems, 2006). We then calculated experimental variograms of both sites for each resolution for each field day with both nearest neighbor and cubic convolution resampled DSMs and fit spherical models to each of these independent experimental variograms using the R package "gstat" (Pebesma, 2004). To estimate the goodness of fit of the spherical models, we calculated RMSE and NMAD values for the fit of the spherical models to the experimental variograms. The maximum distance considered for our variogram calculations was set to one-third of the maximum distance between point pairs within the two scenes, respectively, and we used minimum lag distances equal to the minimum point pair distances. Previous work used one-half of the maximum distance between point pairs as the maximum distance considered for variogram calculations, which would result in the comparison of point pairs at greater lag distances (Schirmer and Lehning, 2011; Clemenzi et al., 2018; Mendoza et al., 2020). Our focus on complex terrain, our relatively small study site extent, and the large number of points to be compared with our high-resolution DSMs motivated our decision for a smaller maximum distance considered for variogram calculations (Blöschl, 1999). Due to the large number of points contained in the high-resolution DSMs (0.02, 0.05, and 0.1 m) and the computing power required for variogram analysis of such large datasets, we used a random sample of 3 million points to process the experimental variograms for these three resolutions. To ensure reproducibility, we used a pre-set seed when randomly sampling. We applied a local polynomial regression (LOESS) fit from the R package 'stats' to the fit spherical models to produce the seasonally averaged resolution specific variograms (R Core Team, 2021). We utilized the United States Geological Survey (USGS) Yeti supercomputer for all of our variogram calculations and model fitting (Falgout and Gordon, 2022).

**3.6 Coefficient of Variation Calculation**

We calculated the coefficient of variation (CV) (Eq. 3) of the vegetation masked and outlier removed snow depth DSMs in the R software environment (R Core Team, 2021) defined as

$$CV = \frac{\sigma}{\mu} \tag{4}$$

where $\sigma$ is standard deviation and $\mu$ is mean snow depth. We calculated the CV values for a variety of nearest neighbor resampled resolutions to ensure consistent results and to reduce the computational load of calculating 0.02 m resolution snow depth DSMs.

## 4 Results

### 4.1 Snow Depth DSM Error and Detrending Results

We compared manual in-situ validation point snow depth measurements (n = 70) with our DSM-differenced snow depths to assess error in our UAS-derived snow depths (Fig. 2). The seasonal mean, standard deviation, RMSE, and NMAD of differences between probed and UAS-derived snow depths show the effect of poor model quality on snow depth measurements (Table 1). The daily mean, standard deviation, and RMSE of differences between UAS-derived and probed snow depths varied considerably throughout the season (Table A2) and showed an increase in accuracy and a slight decrease in precision throughout the season (Fig. A1).

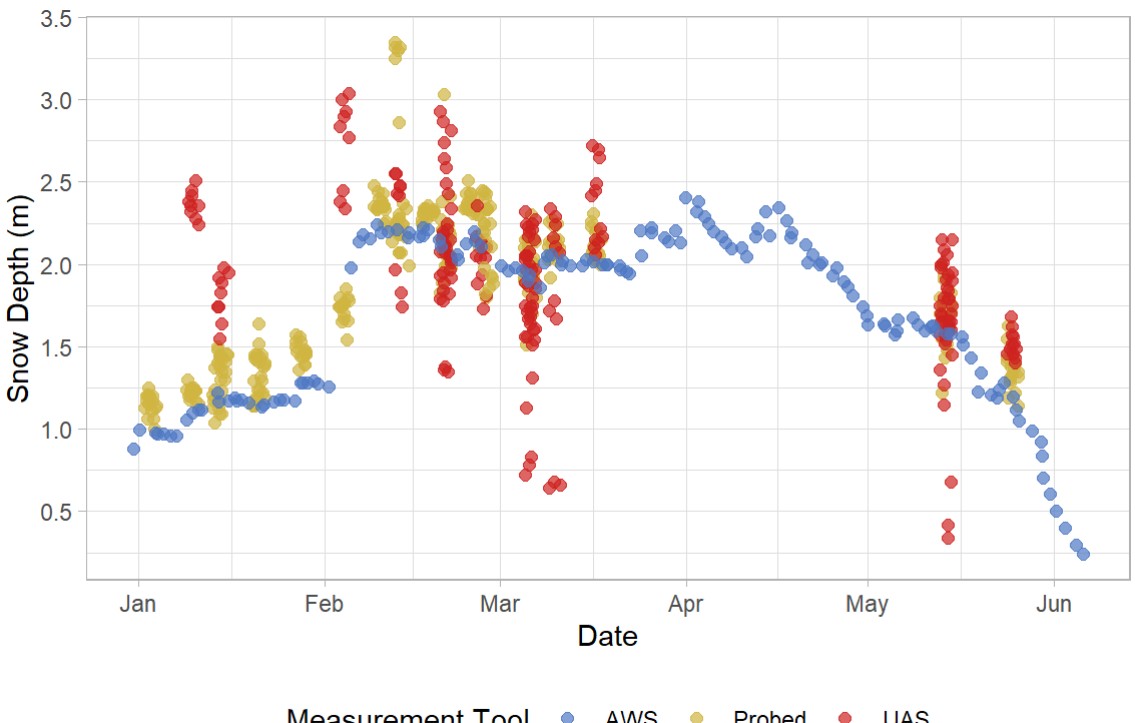

Figure 2: Measured snow depths (m) from 1 January 2020 - 5 June 2020, including additional in-situ probed measurement days where uncrewed aerial system (UAS) models were discarded due to poor surface reconstruction. Brackett meadow automated weather station (AWS, blue) represents the sonic-rangefinder measured snow depth. Probed (yellow) snow depths are collected manually in-situ as validation points. UAS (red) snow depths are derived from digital surface model (DSM)-

differencing at each validation point on a given observation day. Probed points represent snow depth measurements from within the meadow and low relief and lower elevation portions of the Hourglass, and do not represent the same location as the sonic rangefinder attached to the Brackett meadow AWS.


Table 1: Seasonal statistics of observed snow depth differences between probed and uncrewed aerial system (UAS)-derived validation point measurements (DSM = digital surface model; RSME = root mean square error; NMAD = normalized median absolute deviation).

| Snow Depth DSMs included | Mean Difference (m) | Standard Deviation of Differences (m) | RMSE of Differences(m) | NMAD of Differences (m) |
|---|---|---|---|---|
| All | 0.44 | 0.60 | 0.74 | 0.21 |
| Poor Quality Removed | 0.27 | 0.26 | 0.37 | 0.16 |

We collected all validation snow depth measurements, except the crown profile of the 26 February 2020, avalanche, in the lower elevations of the Hourglass and throughout the meadow in order to avoid exposure to snow avalanches. We observed large ranges of measured snow depths within these vegetated areas. For example, the ranges of probed snow depths measured within the 1 m$^2$ validation points (n = 70) were as high as 0.49 m, with an average range of 0.13 m. This assessment is not a comprehensive assessment of error because our validation snow depths were primarily collected at random locations in the
safe lower slopes, and this assessment is therefore biased towards comparisons of measurements in the meadow. Although far from a complete accuracy assessment, our single manual snow depth measurement from upper elevations at the crown of the avalanche (top of the couloir) exhibited a snow depth difference of only 0.01 m (1.90 m measured vs. 1.89 m UAS-derived), which is well within the typical error of manual measurement.

The DSM detrending analysis identified elevation as the most significant independent variable for each day at our study site. Therefore, we detrended all snow depth DSMs using the elevation surface derived from the 8 July 2020 snow free DSM (Fig.3). Complete timeseries plots of vegetation masked and detrended snow depth maps of the Hourglass and meadow are provided in the Appendix (Fig. A7 and A8).

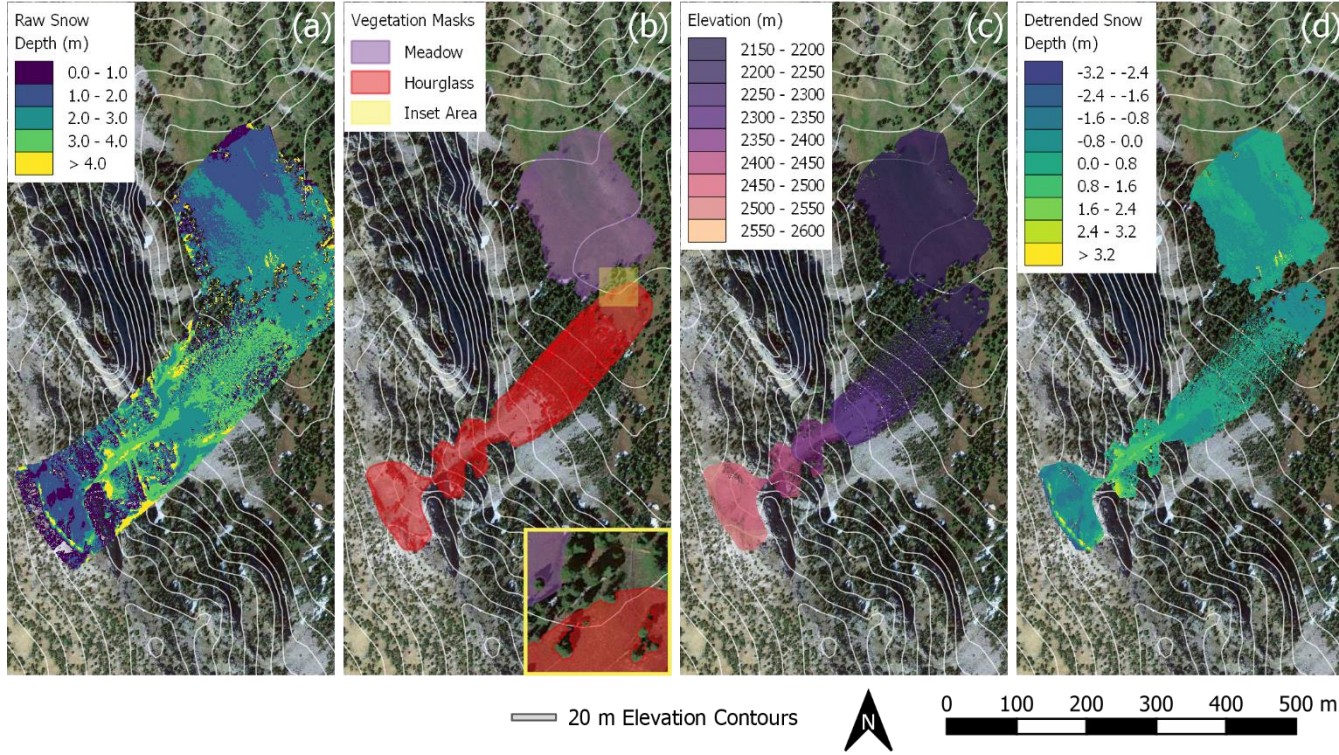

Figure 3: Vegetation masking and detrending process for raw snow depth values from 21 February 2020 derived from DSM-differencing. The scene-wide raw snow depth (a) map is shown prior to vegetation masking (b) and elevation detrending (c), resulting in the detrended snow depth map (d) used for snow depth spatial variability analysis. The inset map (yellow - lower left) of the vegetation mask map (b) illustrates the masks' removal of trees and other vegetation. Elevations (c) are from the 8 July 2022 snow free DSM. Note that detrended snow depths (d) result in negative values at upper elevations. Map satellite imagery: ©Google, ©2022 USDA/FPAC/GEO, Maxar Technologies.

**4.2 Resampling Results**

We resampled and compared the vegetation masked and detrended DSMs from their original 0.02 m spatial resolution to 0.05, 0.1, 0.25, 0.5, 1, 2.5, 5, 10, and 20 m horizontal spatial resolutions using nearest-neighbor, cubic convolution, aggregated mean, and aggregated median methods. We compared the resampled snow depth DSMs by calculating correlation coefficients for all cell values for each pairing of resampling techniques. All resampling techniques are highly correlated at DSM resolutions finer than 1 m, with average correlations of 0.99, 0.98, and 0.97 for 0.25, 0.5, and 1 m resolutions, respectively (Fig. A2). At each resolution step greater than 1 m, correlations between the nearest neighbor, cubic convolution, aggregated mean, and aggregated median methods decrease differentially between the Hourglass and the Meadow but remain consistent between resampling methods. In the Hourglass, average correlations decrease to 0.92, 0.87, 0.77, and 0.73 for 2.5, 5, 10, and

20 m resolutions, respectively. In the Meadow average correlations decrease to 0.94, 0.92, 0.91, and 0.87 for 2.5, 5, 10, and 20 m resolutions, respectively.

### 4.3 Variogram Results

We calculated experimental variograms (Fig. A3) and compared the results of the fitted spherical variogram models from the two distinct topographies within our study site and each field day at a variety of spatial resolutions using two different
resampling techniques, nearest neighbor and cubic convolution. The two resampling techniques produced similar experimental variograms with only a few instances of lower residual semivariance observed in the cubic convolution resampled data. Subtle differences in the experimental variograms between the two resampling techniques influenced the spherical fit models (Fig. A4), The cubic convolution approach fails to register the initial sill break point (around 15 m) of the experimental variogram and fit a larger range with an associated larger sill value in several spherical fit variograms (Fig. A5). The RMSE and NMAD
values for the spherical fit models were consistently higher for the Hourglass than the meadow with the highest values found at spatial resolutions finer than 0.5m and at 20 m (Table A5). We found consistent differences in the range, nugget, and sill (semivariance) values of the Hourglass and the meadow sites. The Hourglass exhibits a smaller range of autocorrelation, greater sill values and greater nugget values than the meadow, when including all dates and snow depth DSM resolutions (Table 2). Specifically, the Hourglass exhibited consistently more snow depth spatial variability on individual field days (Fig.
4) and more seasonal variability in its patterns of spatial variability than the meadow (Fig. 5). These results reflect the given substratum of the two sites. The meadow's more homogenous ground cover and topography are reflected in less variability overall and spatial autocorrelation over greater distances. In contrast, the steep rocky terrain of the Hourglass is reflected in the more dynamic seasonal patterns of spatial variability and shorter distances of autocorrelation. The 20 m resolution variograms frequently misrepresent the spatial variability patterns of finer resolutions and this is perhaps due to the relatively
small study sites creating far fewer point pairs of snow depths to calculate the variograms from, therefore being less resistant to outliers.

Table 2: Average spherical fit variogram results for the Hourglass and meadow for all resolutions using the nearest neighbor resampling method. Mean values from all analysis dates for each location at each resolution.

| Resolution | Hourglass mean range | Meadow mean range | Hourglass mean sill | Meadow mean sill | Hourglass mean nugget | Meadow mean nugget |
|---|---|---|---|---|---|---|
| 0.02 | 10.08 | 46.73 | 0.64 | 0.17 | 0.03 | 0.01 |
| 0.05 | 10.46 | 46.66 | 0.64 | 0.18 | 0.04 | 0.01 |
| 0.1 | 11.30 | 52.71 | 0.63 | 0.17 | 0.06 | 0.02 |
| 0.25 | 12.38 | 57.27 | 0.61 | 0.17 | 0.08 | 0.02 |
| 0.5 | 15.17 | 62.11 | 0.60 | 0.17 | 0.11 | 0.03 |

| 1 | 28.92 | 67.22 | 0.58 | 0.17 | 0.17 | 0.03 |
|---|---|---|---|---|---|---|
| 2.5 | 48.42 | 74.20 | 0.62 | 0.17 | 0.23 | 0.04 |
| 5 | 102.15 | 84.67 | 0.37 | 0.17 | 0.50 | 0.03 |
| 10 | 130.65 | 98.92 | 0.36 | 0.16 | 0.53 | 0.03 |
| 20 | 228.44 | 40.55 | 1.17 | 0.16 | 0.54 | 0.07 |

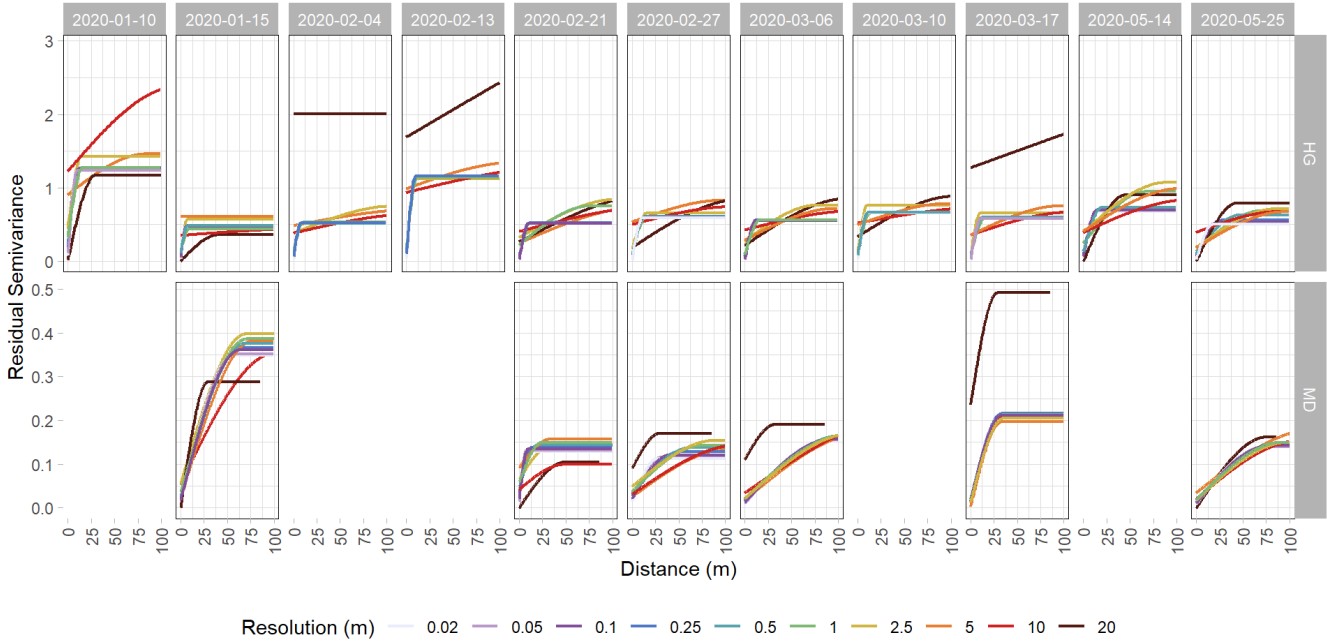

Figure 4: Spherical fit variogram models of detrended snow depth residuals from the Hourglass (HG) and the meadow (MD) using the nearest neighbor resampling method. Each panel depicts a specific observation day and colors represent different snow depth DSM resolutions. Five observation days were removed from the meadow site timeseries due to poor model quality. Note different y-axis scales for the HG and MD rows.

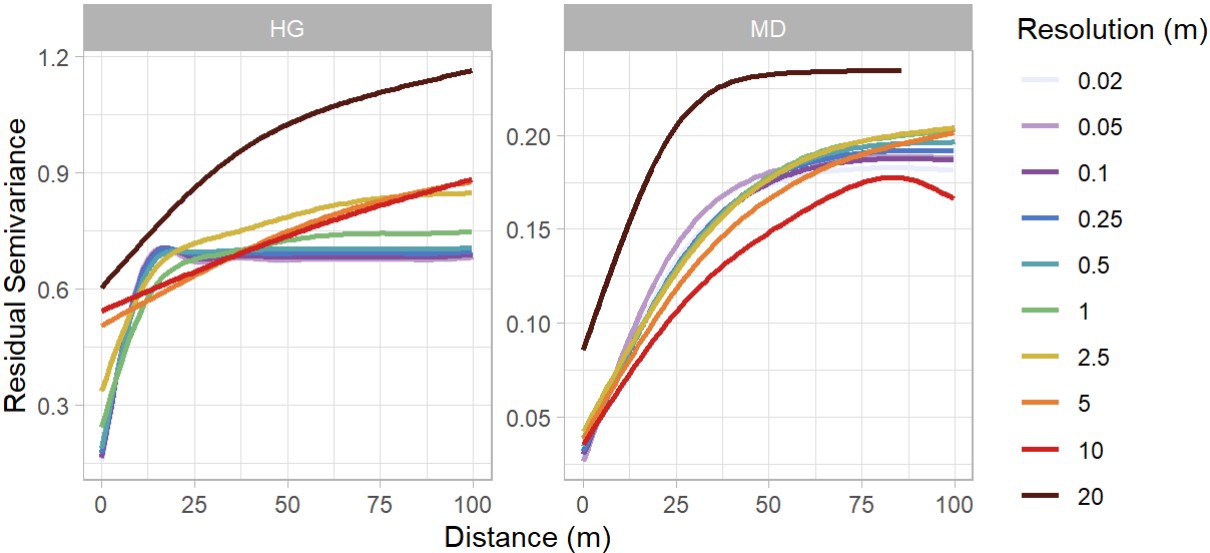

Figure 5: Seasonally averaged, spherical fit variogram models from the Hourglass (HG) and meadow (MD) using the nearest neighbor resampling method. Colors represent different resolutions. Note different y-axis scales.

Temporally, the greatest semivariance values exist earlier in the winter at both the Hourglass and meadow at all resolutions. Variability then decreases throughout mid-winter and increases slightly after the first substantial spring melt event that occurred approximately 2 weeks prior to 14 May 2020. Autocorrelation range generally increased within both the Hourglass and the meadow throughout the winter (Fig. 6) followed by pronounced increases in the spring. Sill values were consistently greater at the Hourglass compared to the meadow throughout the season (Fig. 6) and were relatively similar across all resolutions except 20 m at the Hourglass couloir where greater variability exists throughout the season. Temporally, sill values generally decreased at the Hourglass couloir site and remained consistent at the meadow throughout the winter. Additionally, at the Hourglass, the sill increased at finer resolutions because of a large natural avalanche on 26 February 2020.

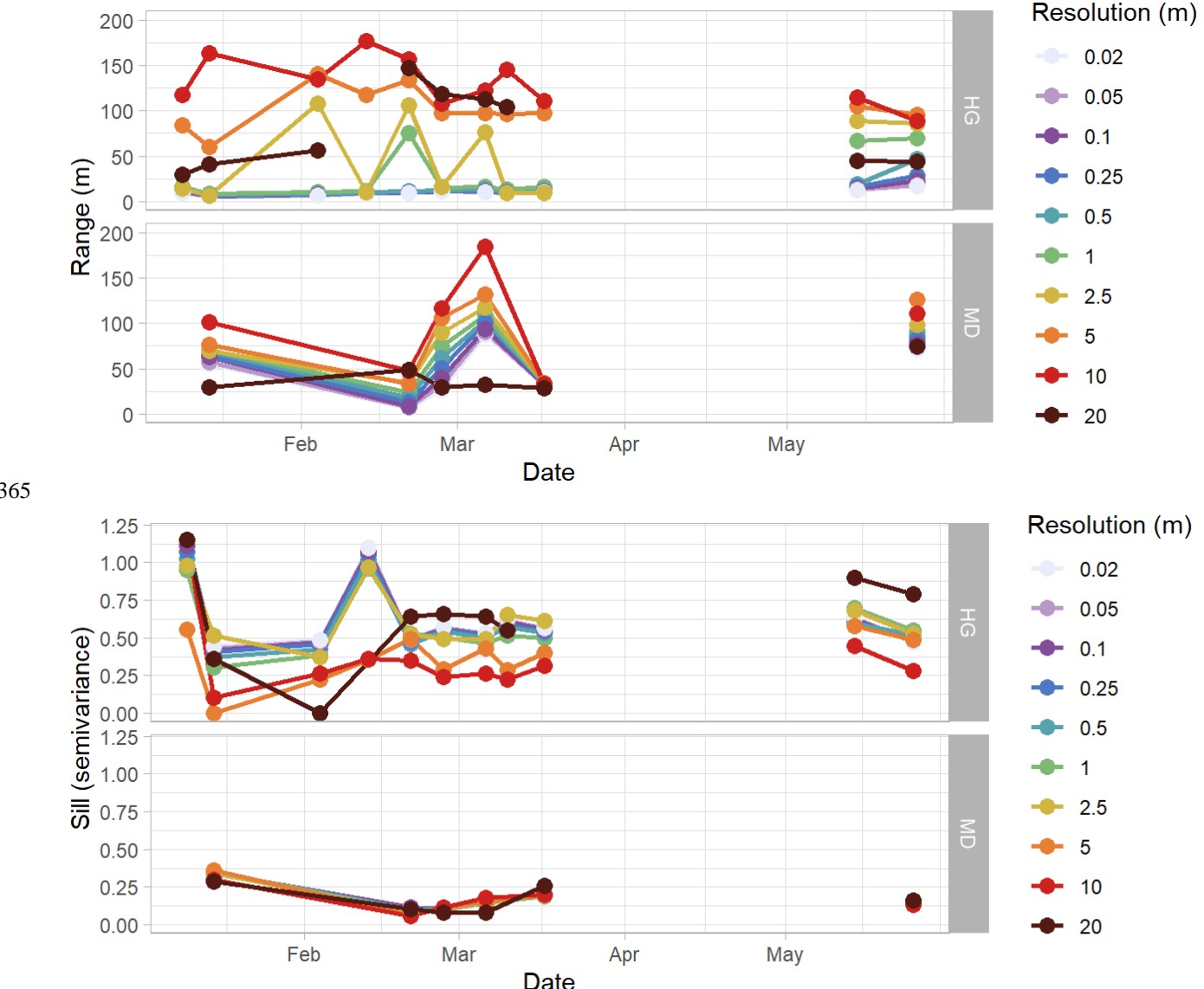


Figure 6: Relative range and sill patterns of spherical fit variogram models of detrended snow depth residuals using the nearest neighbor resampling method in the Hourglass (HG) and meadow (MD) throughout the winter. Colors represent different snow depth digital surface model (DSM) resolutions. The range plot excludes the 20 m range values from the Hourglass for 13

February 2020 and 17 March 2020, which are greater than 200 m.

Snow depth DSM spatial resolution affected the calculated variograms resulting in larger autocorrelation range, sill, and nugget values present in coarser resolution variograms (Fig. 7). At the Hourglass, 0.5 m resolution models accurately represented the spherical fit variograms of all finer resolutions (0.02, 0.05, 0.1, and 0.25 m) and consistently resulted in autocorrelation range

values of ~10 m. 1 m resolution snow depth DSMs captured the pattern of the finer resolution variograms on all but 3 of the
observation dates (21 February, 14 May and 25 May) but aligned more closely with coarser resolution variograms (2.5, 5, 10,
and 20 m) on those 3 observation dates and exhibited larger, more variable autocorrelation ranges consistently throughout the
winter. At the meadow, 2.5 m resolution models accurately represented the spherical fit variograms of all finer resolutions
(0.02, 0.05, 0.1, 0.5, and 1 m) on all observation dates. The 5 m resolution differed from finer resolution patterns when

seasonally averaged, and the 10 m resolution differed from the patterns on a single date and more so when seasonally averaged
(Fig. 5). Snow depth DSM resolutions of 20 m in both topographies and all observation dates failed to capture the patterns of
spatial variability found in finer resolutions.

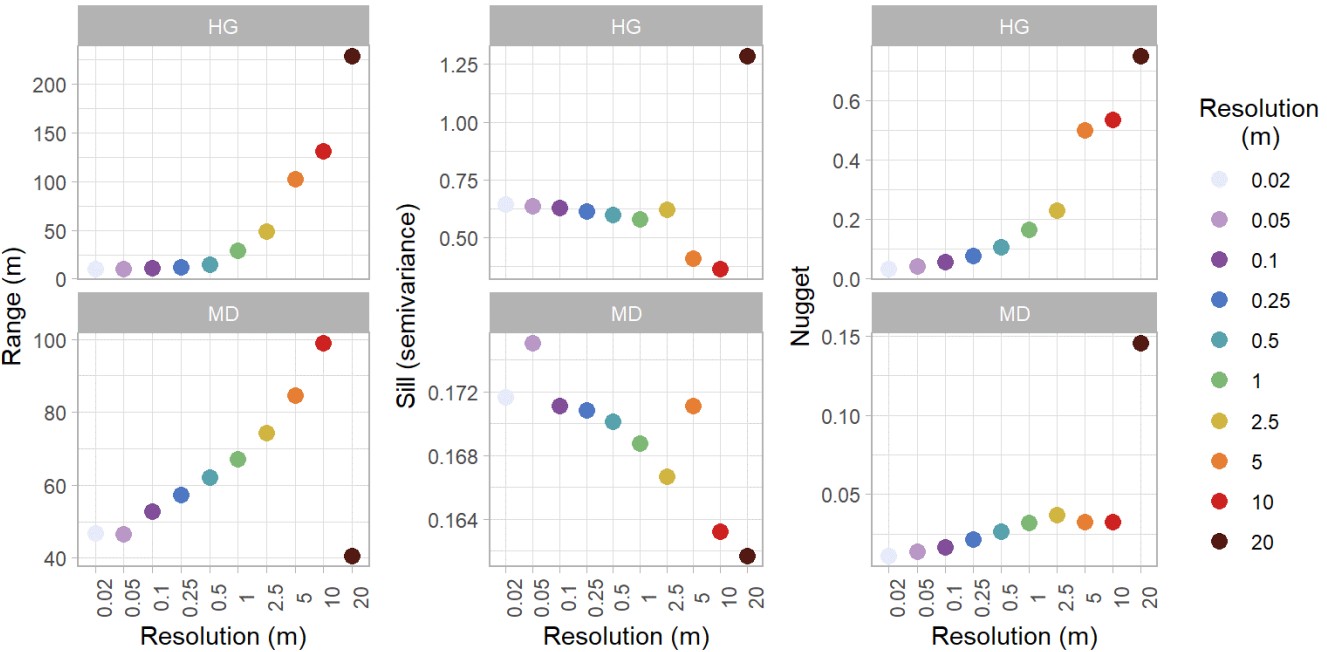

Figure 7: Seasonally averaged range, sill, and nugget values for each resolution from spherical fit models in the Hourglass
(HG) and meadow (MD) using the nearest neighbor resampling method. Colors represent different snow depth digital surface
model (DSM) resolutions

### 4.4 Coefficient of Variation Results

We compared the calculated coefficient of variation over time at the Hourglass and the meadow. Resultant coefficients of

variation were similar across a variety of snow depth DSM resolutions. As such we present the 0.5 m resolution results here.
Coefficient of variation values were greater at the Hourglass when compared to the meadow on every observation day (Fig.
8). The seasonal pattern of variability in the Hourglass started higher in January, decreased then remained consistent through

March, and peaked in May concurrently with the onset of ablation. At the meadow, the variability decreased throughout the season before peaking in May with the onset of ablation.

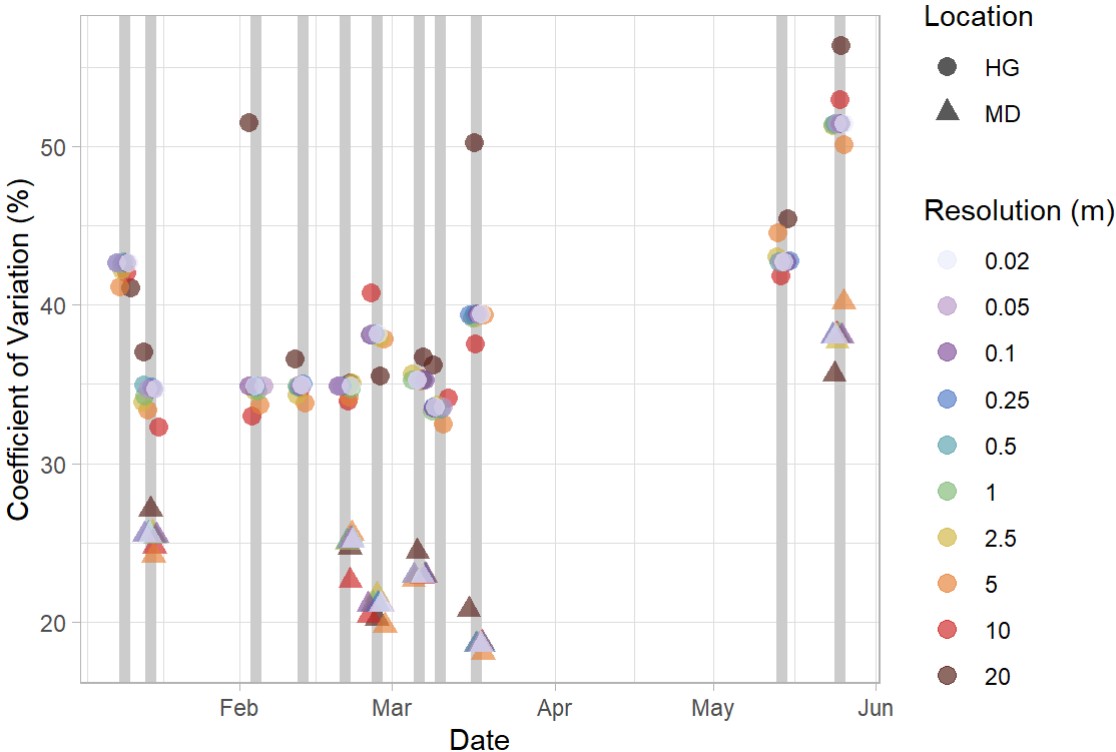

Figure 8: Coefficient of variation values (%) for each observation day at the Hourglass (HG, circles) and meadow (MD, triangles) as calculated from all resolutions (m) of snow depth digital surface models using the nearest neighbor resampling method. The points are slightly scattered horizontally around the collection dates (grey vertical lines) to allow for clearer viewing and interpretation. Colors represent different snow depth digital surface model (DSM) resolutions.

## 5 Discussion

In this study, we analyzed a time series of 11 high-resolution snow depth DSMs derived from UAS and SfM techniques in a 0.2 km² study area containing steep-complex and protected-simple terrain in the Bridger Range of Montana, USA. We collected these data to investigate the scales of spatial variability of snow depth in complex mountain terrain, compare with the spatial variability observed in adjacent simple mountain terrain, and explore the temporal evolution of spatial variability patterns of snow depth.

### 5.1 Snow Depth Differences and Detrending

Comparisons between DSM-differenced and probed snow depths highlight the challenge of sampling spatially representative snow depth measurements with underlying vegetation. The 1 m$^2$ validation point measurements had an average range of 0.13 m while the mean difference between DSM-differenced snow depths and in-situ probed snow depths was 0.27 m (Table 1). Both of these point and observational tool measurement differences, as well as our additional error estimates, could be attributed to the vegetation captured in the snow-free 8 July 2020 DSM. This vegetated surface had greater than 0.5 m of vertical variability across horizontal distances less than 1 m and compressed at an unquantified and spatially heterogenous rate under the gradually increasing snowpack. This vegetation effect is largely confined to the lower elevations of our study site, which is also primarily where we collected our validation point measurements. Vegetation effect is a recognized weakness of UAS-derived snow depth measurement and helps explain the differences we observed between the DSM-differenced snow depths and probed validation point measurements in this study (Bühler et al., 2016).

Previous research identified wind direction as a contributing variable to their spatial variability findings (Clemenzi et al., 2018; Deems et al., 2006; Mendoza et al., 2020; Mott and Lehning, 2010; Mott et al., 2018). Our results suggest that elevation is the most significant predictor in directional snow distribution bias in our dataset and we detrended the DSMs on this metric.

### 5.2 DSM Resampling Methods

Our project analyzed the spatial variability of snow depth across spatial resolutions spanning three orders of magnitude (0.02 m to 20 m). Given this large resampling need, we were interested in the effects of resampling technique on the resultant spatial variability. Previous research identifies over-smoothing as a concern with resampling methods which rely on averaging because it results in decreasing the absolute magnitude of variance observed (Fassnacht and Deems, 2006; Melvod and Skaugen, 2013). We found very high correlation of residual snow depths (Sec. 4.2) between all resampling techniques at spatial resolutions finer than 1 m (Fig. A2). As resolution increased beyond 1 m, correlation begins to decrease, especially in the more heterogenous terrain of the Hourglass. Closer inspection of cell-by-cell differences reveals the cubic convolution and aggregated mean methods producing unrealistic snow depth residual artifacts near areas of greater snow depth variability, such as the avalanche crown, near cornices, and in the upper start zone. This is probably due to these resampling techniques' limited resistance to outliers. Additionally, we observed very similar patterns in the experimental variograms between nearest neighbor and cubic convolutions methods (Fig. A3). Consistently slightly lower semivariance values in cubic convolution resampled experimental variograms point towards potential over-smoothing of the natural variability seen in the nearest neighbor resampled experimental variograms. On the other hand, subtle differences in the cubic convolution experimental variograms propagated larger differences in the spherically fit models (Fig. A4) and resulted in bother greater ranges of autocorrelation and higher semivariance values (Fig. A5 and A6). The range and semivariance values observed at 0.25, 0.5, and 1 m resolutions resembled those found at 2.5 and 5 m resolutions in the nearest neighbor spherically fit models. As spatial resolutions coarsen,

averaging resampling methods produce longer ranges (Fassnacht and Deems, 2006) and, with that, higher sill values. We are confident that nearest neighbor resampling methods depict the true patterns of spatial variability of snow depth at fine
resolutions within our study area because of the preservation of real observed snow depth values and the high correlation to other resampling methods. Given the diverging results at resolutions greater than 1 m, we urge careful consideration of resampling techniques for coarser spatial resolutions in future work.

### 5.3 Scales of Spatial Variability

Many previous studies utilized 1 m resolution sampling grids (Clemenzi et al., 2018; De Michele et al., 2016; Deems et al.,
2013; López-Moreno et al., 2015; Mendoza et al., 2020; Meyer and Skiles, 2019; Trujillo et al., 2009) but without detailed analysis to determine if this resolution is sufficient to capture the patterns of spatial variability. Our results indicate 1 m resolution is an insufficient resolution to capture the complete pattern of snow depth spatial variability in the steep complex mountain terrain of the Hourglass. At our site, 0.5 m resolution results capture the spatial variability patterns seen in all finer resolutions (0.02, 0.05, 0.1, and 0.25 m) in each observation day at the Hourglass (Fig. 5), while the 1 m resolution only
captures these patterns on seven of our eleven observation days. Maximum variation exists within a 15 m range for all sub-0.5 m resolution variograms with a decreasing variance as the range continues to grow beyond the sill. Coarser resolution variograms exhibit increasing variance at greater ranges, with increasing variance beyond the sill. While 1 m, and even 2.5 m, resolutions have similar ranges as finer resolution models on some observation days, the mean range values increase dramatically between 0.5 m (15 m), 1 m (29 m), and 2.5 m (48 m) resolutions and decrease minimally below 0.5 m resolutions
(10 to 12 m for 0.02 and 0.25 m resolutions, respectively) (Table 2 and Fig. 7). Our results suggest that a 0.5 m sampling resolution is the coarsest sampling resolution necessary to capture all small-scale spatial variability of snow depth in the complex mountain terrain of our study site.

However, our results suggest 2.5 m resolution sampling grids are adequate to capture spatial variability in the protected terrain
at the meadow. The finer resolution patterns evident in the mean variograms of the meadow are similar to the 2.5 m resolution while the 5, 10 and 20 m resolutions differ distinctly with generally larger range values (Fig. 5). The autocorrelation range values at the meadow scale directly with snow depth DSM resolution while the sill values remain consistent across snow depth DSM resolutions (Fig. 7). Given this relationship, the 2.5 m resolution captures both the fine and coarse resolution patterns in the meadow. This distinct difference in snow depth spatial variability between complex and simple terrain provides evidence
of the contrasting snow distributions in the two mountain topographies.

Previous research reported autocorrelation range values for snow depths of 15 - 25 m in a variety of mountain terrain with an additional correlated scaling break above 50 m (Fassnacht and Deems, 2006; Clemenzi et al., 2018; López-Moreno et al., 2015; Mendoza et al., 2020; Trujillo et al., 2009). We found consistently less than 20 m range values in the steep complex terrain in
the Hourglass at finer spatial resolutions and greater than 50 m range values in the meadow at all resolutions (Table 2). Our

results also suggest that the range of autocorrelation increased throughout the winter at both sites. We attribute this to increasingly homogenous snow depth distributions over larger distances due to wind redistribution near ridges (Mott and Lehning, 2010; Trujillo et al., 2007) and small-scale redistribution processes (Mott et al., 2011). Increasingly leptokurtic distributions evident in scene-wide violin plots, especially at the meadow, indicate that snow depth distributions were largely
concentrated near the mean snow depth for each given observation day (Fig. A9) and became more uniform as snow depth increases.

Our results show that snow depth variability generally decreased throughout the season in the complex terrain of the Hourglass (Fig. 8). This suggests that a deeper snowpack tends to decrease the spatial variability of snow depth as terrain and vegetation
features become less influential on snow depth distribution across space (Deems et al., 2006; Elder et al., 1991; Harder et al., 2016; Trujillo et al., 2009). There was a slight increase in snow depth variability at the Hourglass following two notable events: a natural avalanche on February 26 and the first major spring melt during the first half of May (Fig. 4). An increase in sill values and a clear distribution change of snow depth residuals provide evidence for these snow depth distribution shifts (Fig. A9). The spring-melt-aligned increase in spatial variability is similar to changes during ablation periods reported by López-
Moreno et al. (2015) who used the coefficient of variation as the measure of variability over the course of eight observation days spread over two winters.

### 5.4 Limitations

Snow depth DSM creation through UAS and SfM photogrammetry workflows is distinctly challenged by snow surface conditions and their interaction with local lighting (Bühler et al., 2016, 2017; Goetz and Brenning, 2019). DSMs from certain
observation days had to be removed from further analysis due to poor quality, which can be attributed to homogenous snow surfaces that occurred due to either recent snowfall with minimal wind, cloud cover producing low light, or a combination of both. The wind-sheltered and nearly flat terrain of the meadow limited the influence of surface texture creating processes, such as wind-redistribution and natural snow sluffing, resulting in uniform minimally textured snow surfaces and more observed days removed from further analysis (Table A3). These poor-quality models affected errors in snow depth measurement (Table
1, Table A4) and, once removed, the error in our remaining models was comparable to the reported error margins of other similar UAS-derived snow depth research (Adams et al., 2018; Eberhard et al., 2021; Revuelto et al., 2021). Therefore, we are confident that our retained UAS-derived snow depth observations in steep mountain terrain are accurate.

The analytical approach used in this study is limited by computational resource availability. The processing time for variogram
analysis scaled directly with snow depth DSM resolutions (Fig. A10) and increased exponentially at finer resolutions while processing in parallel on the USGS Yeti supercomputer (specifications online). We utilized a simple random sample of 3 million points for all 0.02 and 0.05 m resolution DSMs to avoid memory overloading. Processing times for high-resolution

DSM analysis at resolutions finer than 0.5 m offered little additional value (see Figs. 5, 6, and 7) given the computational requirements thereby supporting a spatial sampling grid of 0.5 m for snow depth spatial variability analysis.


Our results are from a thorough analysis of a single study site and under the influence of only the interactions of the local topography and meteorological events from one winter season. The spatially limited in-situ snow depth validation measurements are not completely representative of our entire research site, particularly at upper elevations near the ridgeline. Additionally, our snow depth measurement errors from the UAS-SfM photogrammetry process may contribute to snow depth

spatial variability error, but these errors are challenging to accurately quantify (Redpath et al., 2018) and likely contribute a trivial amount (Adams et al., 2018; Eberhard et al., 2021; Revuelto et al., 2021). We also attempted to account for these small errors by conducting repeated UAV flights over the course of a season, using GCPs from the same locations on both snow-free and snow-covered sampling flights, and choosing a site with a relatively deep snowpack (Goetz and Brenning, 2019). Future snow depth spatial variability research should consider observing a wider variety of complex mountain terrain features,

different snow and avalanche climates, and using additional remote sensing tools for further validation or comparison.

**6 Conclusions**

This study quantifies the relevant spatial sampling scales for accurately mapping snow depth spatial variability in the complex mountain terrain of a study site in the Bridger Range of Montana, USA. We used a time series of uncrewed aerial systems (UAS)-derived centimeter-scale models of evolving snow distribution in a steep complex couloir as well as an adjacent

sheltered flat mountain meadow. Our results suggest that a nearest neighbor resampling technique maintains the naturally occurring spatial variability of snow depths at spatial resolutions of 1 m or finer. We demonstrate that 0.5 m sample spacing resolution is necessary for accurately capturing the naturally occurring spatial variability of snow depth in complex terrain at our study site. This finding contrasts with previous research that typically utilized 1 m resolution models. However, in protected, simple mountain terrain we show that 2.5 m sample spacing is sufficient. This test of extremely fine resolution

surface models is relevant for the planning of future snow depth studies in mountain environments both from a spatial variability and processing perspective. Not only does capturing 0.5 m resolution data increase field efficiency, whether by traditional methods or using remote sensing approaches, but it also decreases the computational expense of processing and analyzing the data. This resolution improves our ability to observe large spatial extents with confidence that accurate measurements of snow depth spatial variability are captured.


We show consistent snow depth autocorrelation ranges to be 10-20 m in steep complex terrain of the Hourglass and 50–65 m in the meadow, which aligns with scaling breaks identified in previous literature on snow depth spatial variability (Deems et al., 2006; López-Moreno et al., 2015; Mendoza et al., 2020). We also show that the steep complex terrain in the Hourglass exhibited greater spatial variability over smaller distances throughout the winter than the protected simple terrain of the

meadow. Additionally, we show that the seasonal evolution of spatial variability is not the same in both topographies. The specific spatial and temporal scales at which snow depth varies within these two terrains influence sampling strategies as they relate to topography and our understanding of snow distributions within the varied mountain landscape we depend on for water resource storage and recreation.

**Appendices**

Table A1: Average locational error for UAS collected imagery after PPK processing. Mean locational difference values are calculated from all images collected and processed on a given field day and have been calculated for x, y, and z (latitude, longitude, and elevation, respectively) directions.

| Date | Post PPK diff x (m) | Post PPK diff y (m) | Post PPK diff z (m) |
|---|---|---|---|
| 10 January 2020 | 0.003 | 0.001 | 0.19 |
| 15 January 2020 | 0.002 | 0.006 | 0.19 |
| 4 February 2020 | 0.006 | 0.0004 | 0.19 |
| 13 February 2020 | 0.004 | 0.001 | 0.19 |
| 21 February 2020 | 0.006 | 0.006 | 0.19 |
| 27 February 2020 | 0.015 | 0.006 | 0.19 |
| 6 March 2020 | 0.005 | 0.003 | 0.19 |
| 10 March 2020 | 0.004 | 0.009 | 0.19 |
| 17 March 2020 | 0.001 | 0.003 | 0.19 |
| 14 May 2020 | 0.0004 | 0.001 | 0.19 |
| 25 May 2020 | 0.005 | 0.004 | 0.19 |
| 8 July 2020 | 0.017 | 0.022 | 0.19 |

Table A2: Models removed due to poor quality.

| Location | Observation Dates Removed From Analysis |
|---|---|
| Hourglass | 7 January 2020 |
| Meadow | 7 January 2020, 10 January 2020, 4 February 2020, 13 February 2020, 10 March 2020, 14 May 2020 |


Table A3. Summary statistics of all measured and uncrewed aerial systems (UAS)-derived snow depths (HS) from the Hourglass and meadow on all sampling days. Low-quality modelled meadow days removed from analysis filled grey. All values reported in meters.

| Sampling day (all 2020) | Brackett Meadow AWS Measured HS (m) | Mean Probed HS (m) | Probed HS Range (m) | UAS-derived mean HS at Validation Points (m) | UAS-derived HS Range (m) | Mean Difference Between Probed HS and UAS-derived HS (m) | RMSE of Difference Between Probed HS and UAS-derived HS (m) | Standard Deviation of Difference Between Probed HS and UAS-derived HS (m) |
|---|---|---|---|---|---|---|---|---|
| 7 January | 0.96 | 1.16 | 0.09 | 1.23 | 0.31 | 0.7 | 0.15 | 0.16 |
| 10 January | 1.11 | 1.22 | 0.13 | 2.36 | 0.1 | 1.14 | 1.14 | 0.05 |
| 15 January | 1.22 | 1.41 | 0.11 | 1.8 | 0.28 | 0.39 | 0.41 | 0.17 |
| 4 February | 1.51 | 1.73 | 0.11 | 2.74 | 0.59 | 1.01 | 1.04 | 0.27 |
| 13 February | 2.22 | 2.59 | 1.11 | 2.28 | 0.63 | 0.32 | 0.86 | 0.98 |
| 21 February | 2.11 | 2.12 | 0.12 | 2.39 | 4.25 | 0.27 | 1.1 | 1.11 |
| 27 February | 2.11 | 2.26 | 0.29 | 2.03 | 0.41 | 0.23 | 0.34 | 0.29 |
| 6 March | 1.90 | 1.95 | 0.79 | 1.79 | 1.51 | 0.16 | 0.37 | 0.35 |
| 10 March | 2.07 | 2.12 | 0.17 | 1.7 | 1.63 | 0.42 | 0.74 | 0.7 |
| 17 March | 2.01 | 2.11 | 0.2 | 2.35 | 0.57 | 0.24 | 0.31 | 0.22 |
| 14 May | 1.57 | 1.75 | 0.65 | 1.92 | 3.42 | 0.31 | 0.81 | 0.77 |
| 25 May | 1.19 | 1.37 | 0.14 | 1.51 | 0.17 | 0.15 | 0.17 | 0.09 |

Table A4: SAGA-GIS DTM Filter tool settings.

| Setting | Values |
|---|---|
| Radius | 10 m |
| Slope | 30° |
| Use Confidence Intervals | Yes |

Table A5: Resolution-averaged root mean squared error (RMSE) and normalized median absolute deviation (NMAD) of spherical fit variogram models of the Hourglass (HG) and meadow (MD). All results are unitless values of semivariance.

| Location | Resolution (m) | Average RMSE | Average NMAD |
|---|---|---|---|
| HG | 0.02 | 0.176218 | 0.16362 |
| HG | 0.05 | 0.174146 | 0.163597 |
| HG | 0.1 | 0.171359 | 0.160636 |
| HG | 0.25 | 0.165605 | 0.156143 |
| HG | 0.5 | 0.158617 | 0.149346 |
| HG | 1 | 0.133527 | 0.125087 |
| HG | 2.5 | 0.10112 | 0.09462 |
| HG | 5 | 0.041416 | 0.04496 |
| HG | 10 | 0.065416 | 0.067764 |
| HG | 20 | 0.181545 | 0.082836 |
| MD | 0.02 | 0.021752 | 0.020394 |
| MD | 0.05 | 0.020563 | 0.01912 |
| MD | 0.1 | 0.018766 | 0.017849 |
| MD | 0.25 | 0.01708 | 0.016117 |
| MD | 0.5 | 0.015251 | 0.016166 |
| MD | 1 | 0.013839 | 0.015157 |
| MD | 2.5 | 0.01285 | 0.01327 |
| MD | 5 | 0.011784 | 0.012584 |
| MD | 10 | 0.009675 | 0.008814 |
| MD | 20 | 0.078682 | 0.026842 |


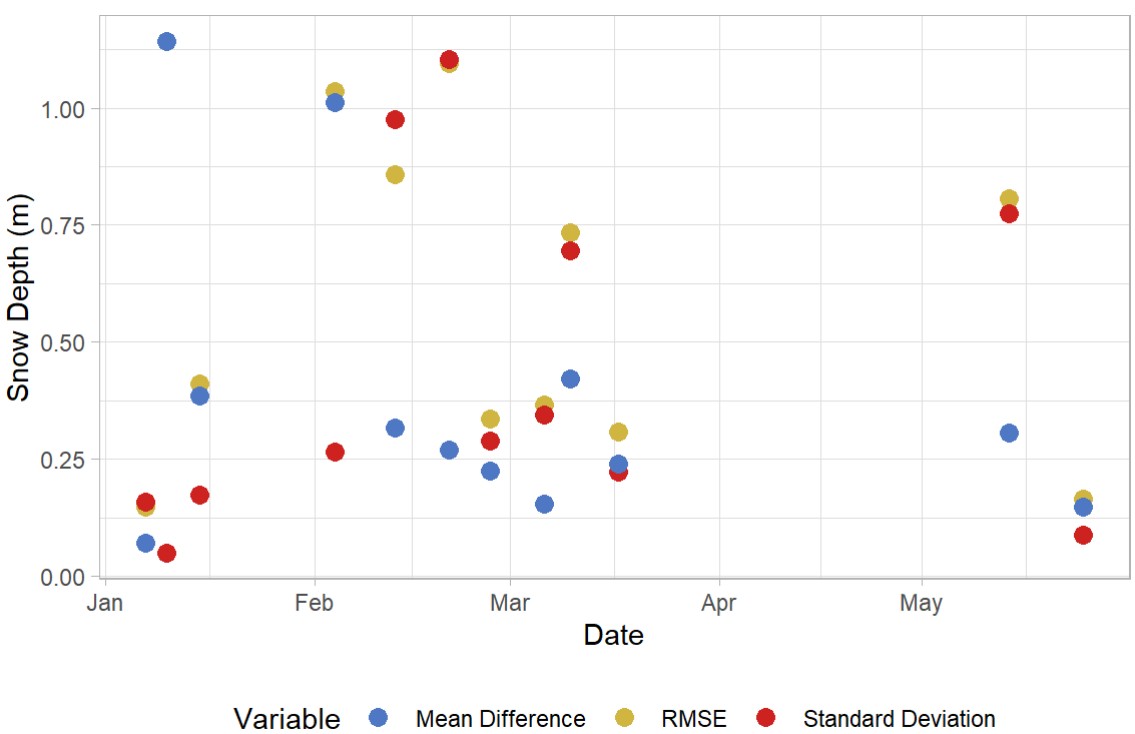

Figure A1: Snow depth uncrewed aerial systems (UAS)-derived error (m) throughout the 2019/2020 winter field season. Each variable is calculated with all manually probed validation points and corresponding UAS-derived snow depths for each observation day.

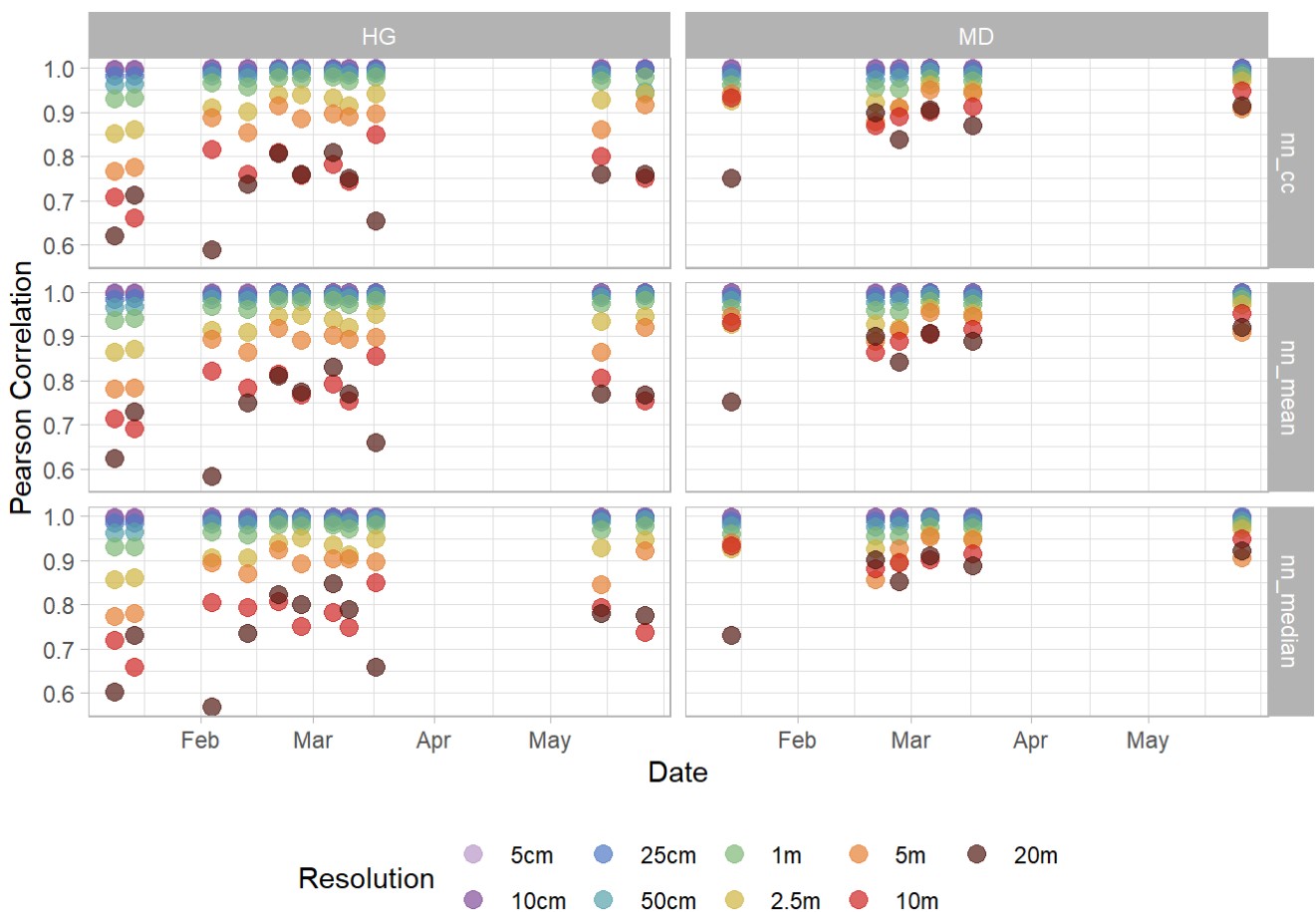

Figure A2: Pearson correlations of nearest neighbor (NN), cubic convolution (CC), aggregated mean (mean), and aggregated median (median) resampled snow depth residuals for the Hourglass (HG) and meadow (MD) locations for each field day. Each row represents nearest neighbor correlations with cubic convolution (top), aggregated mean (middle) and aggregated median (bottom). Colors represent different snow depth DSM resolutions.

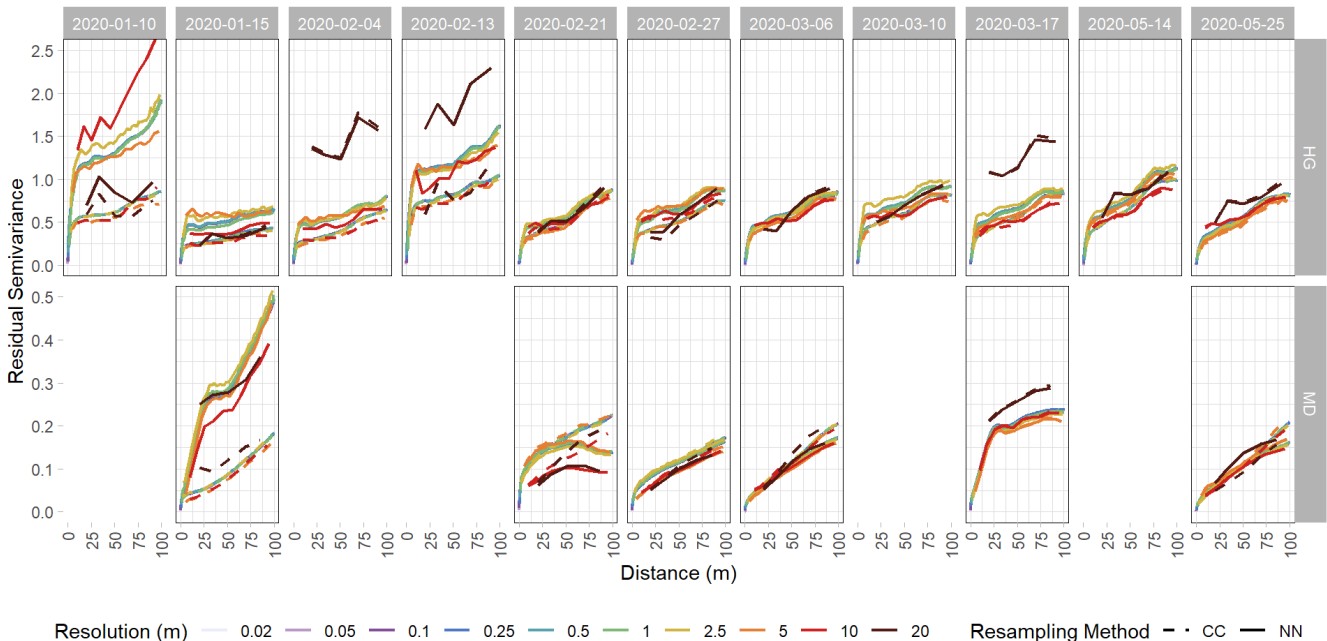

Figure A3: Experimental variograms of detrended snow depth residuals from the Hourglass (HG) and the meadow (MD). Each panel depicts a specific observation day, with solid lines representing nearest neighbor (NN) resampling and dashed lines representing cubic convolution (CC) resampling methods, and colors represent different snow depth DSM resolutions. Five observation days were removed from the meadow site timeseries due to poor model quality. Note different y-axis scales for the HG and MD rows.

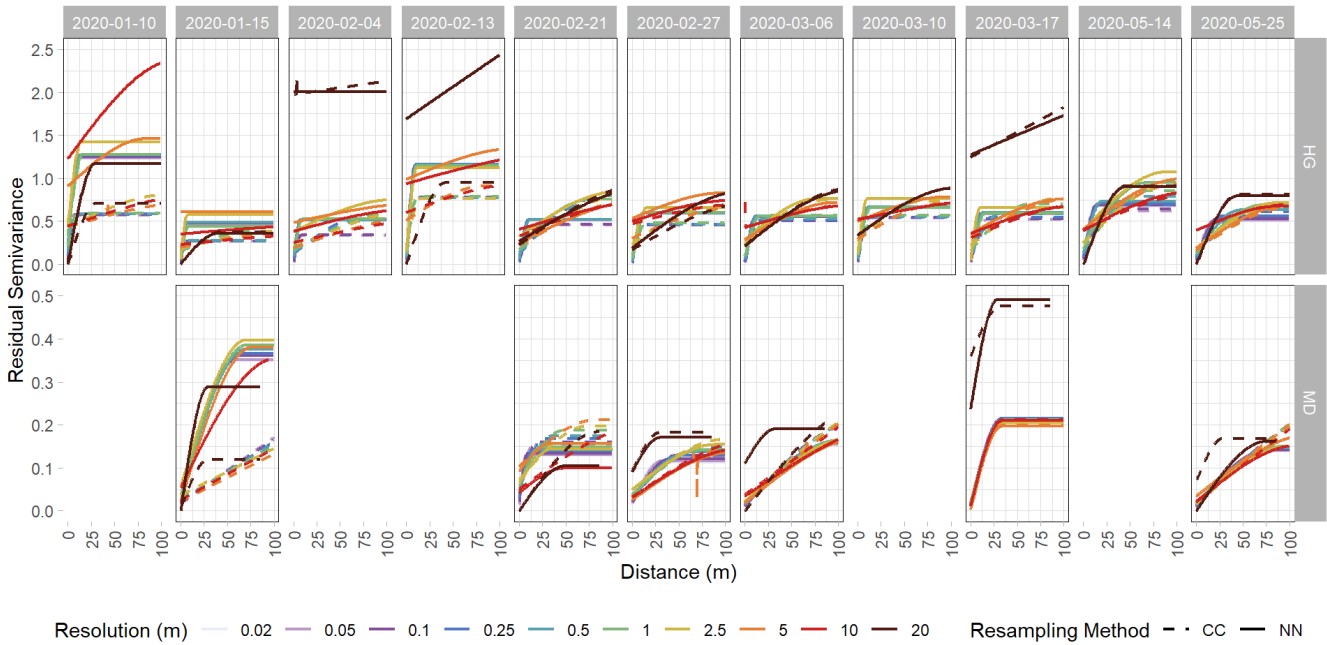

Figure A4: Spherically fit variograms of detrended snow depth residuals from the Hourglass (HG) and the meadow (MD). Each panel depicts a specific observation day, with solid lines representing nearest neighbor (NN) resampling and dashed lines representing cubic convolution (CC) resampling methods, and colors represent different snow depth DSM resolutions. Five observation days were removed from the meadow site timeseries due to poor model quality. Note different y-axis scales for the HG and MD rows.

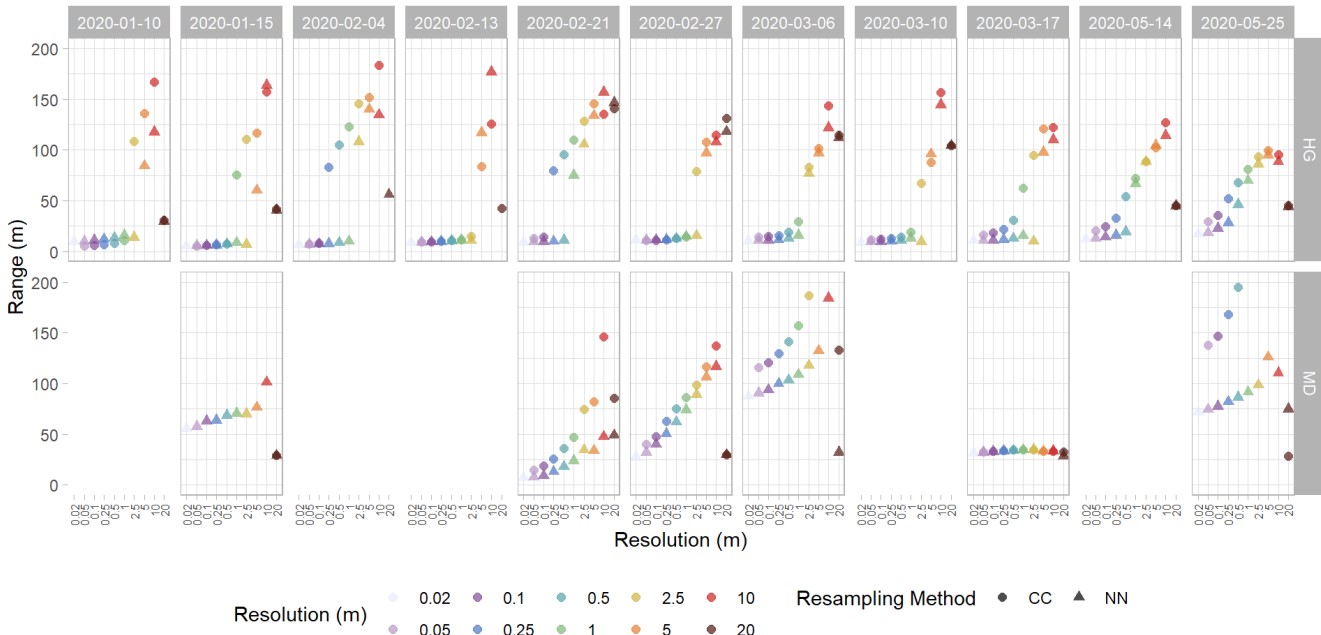

Figure A5: Range values from spherically fit variograms of detrended snow depth residuals from the Hourglass (HG) and the meadow (MD). Each panel depicts a specific observation day, with circles representing cubic convolution (CC) resampling and triangles representing nearest neighbor (NN) resampling methods, and colors represent different snow depth DSM resolutions. Five observation days were removed from the meadow site timeseries due to poor model quality.

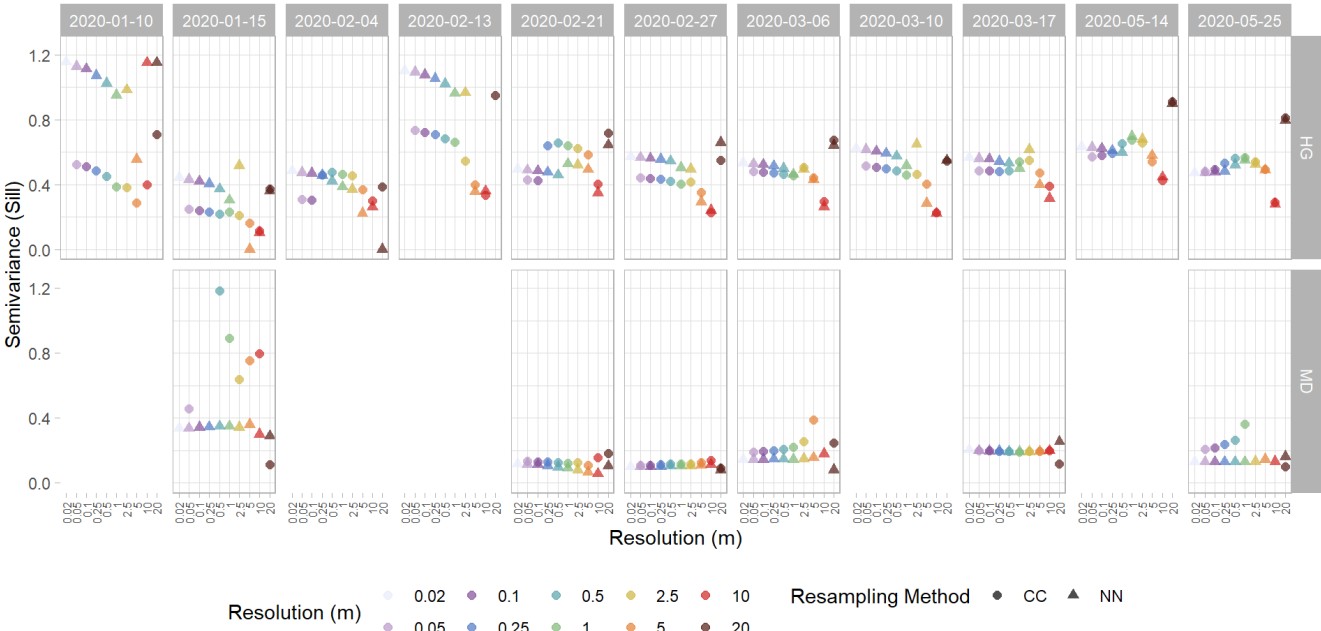

Figure A6: Sill (semivariance) values from spherically fit variograms of detrended snow depth residuals from the Hourglass
(HG) and the meadow (MD). Each panel depicts a specific observation day, with circles representing cubic convolution (CC)
resampling and triangles representing nearest neighbor (NN) resampling methods, and colors represent different snow depth
DSM resolutions. Five observation days were removed from the meadow site timeseries due to poor model quality.

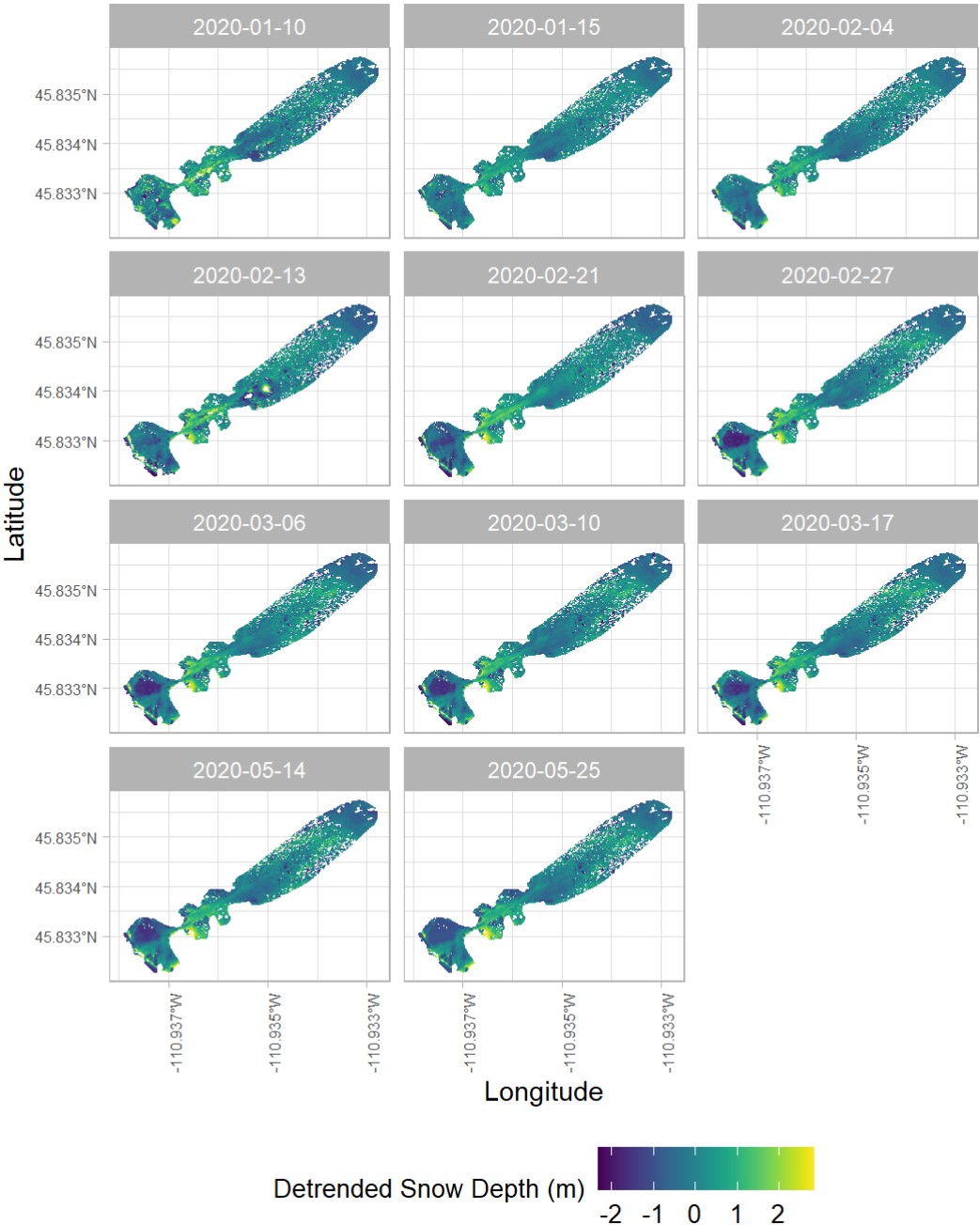

Figure A7: Timeseries of detrended snow depth maps of the Hourglass study site.

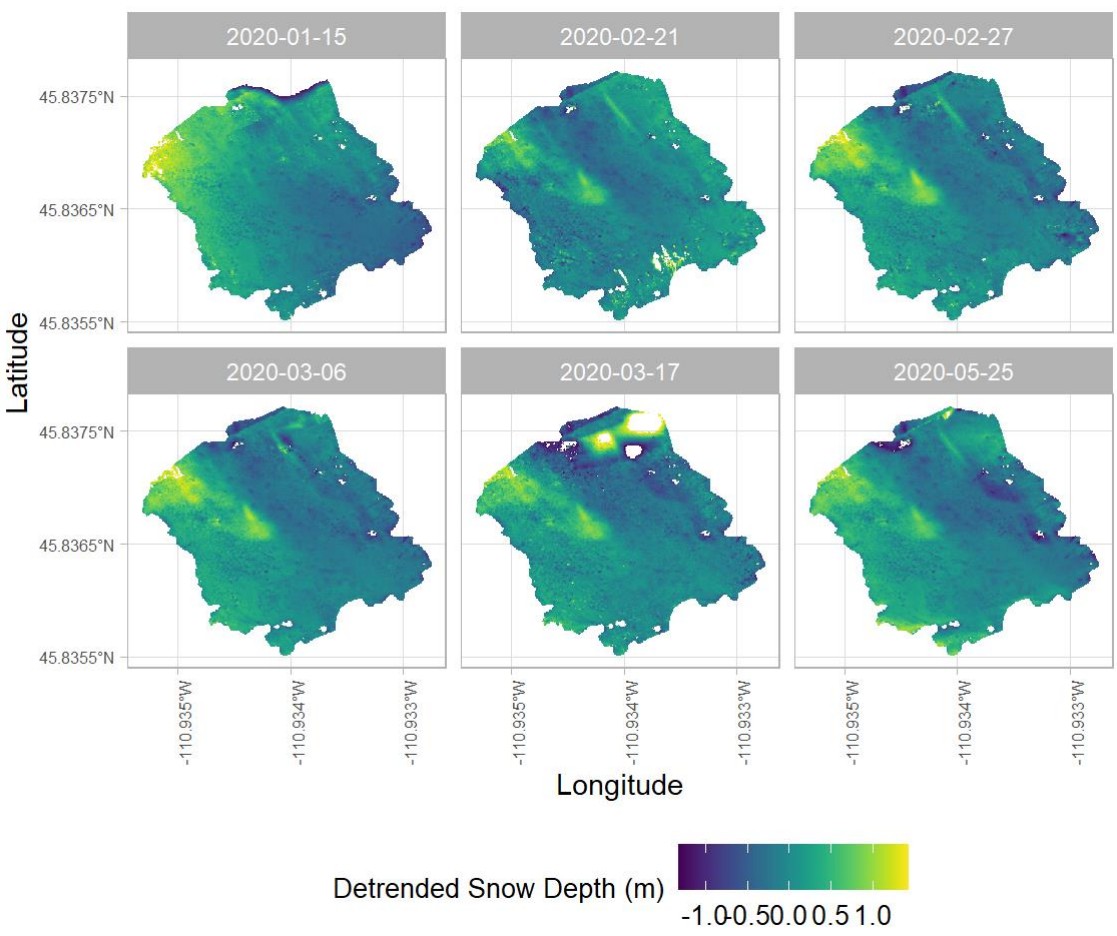

Figure A8: Timeseries of detrended snow depth maps of the meadow study site.

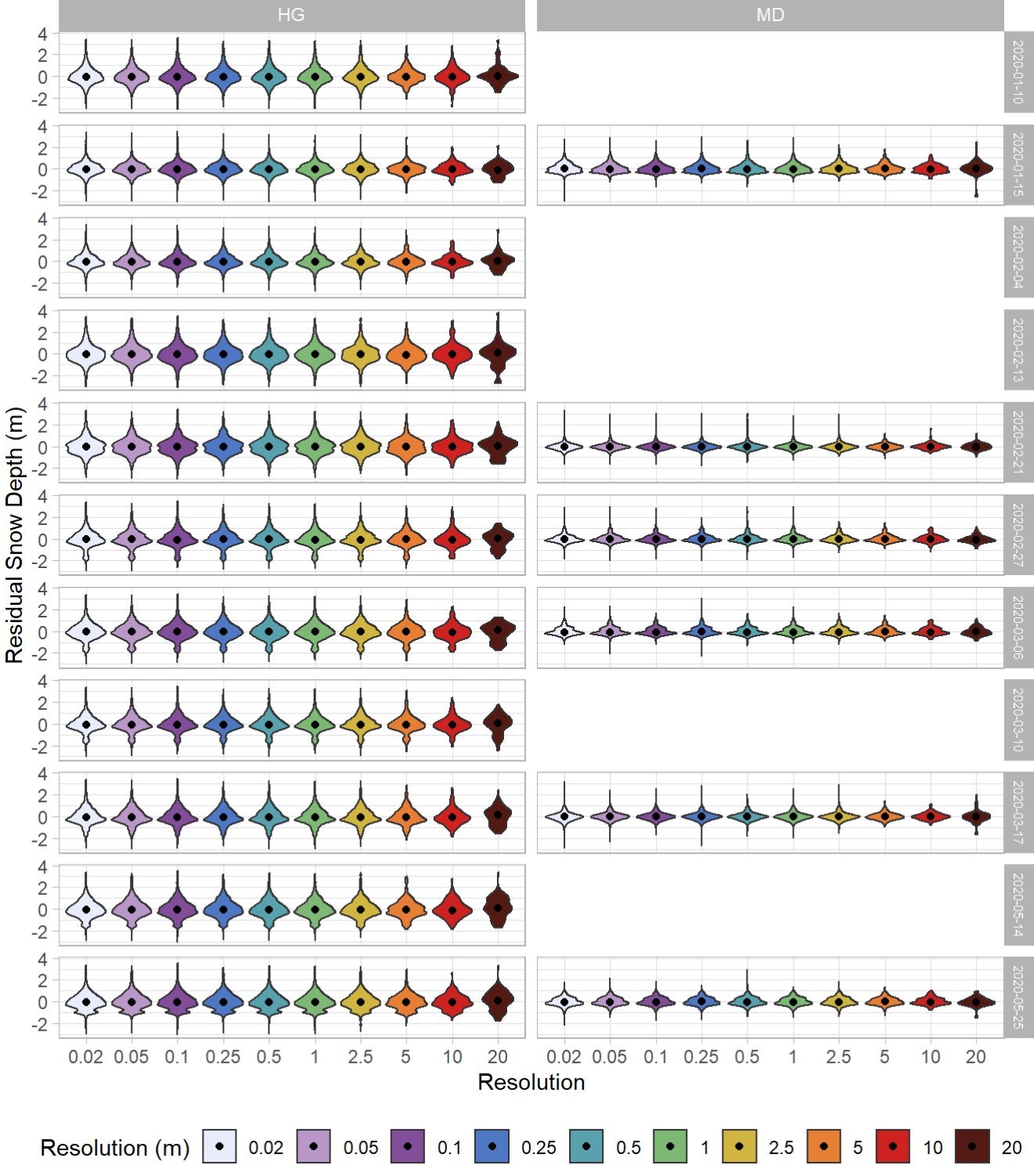

Figure A9: Violin plots of detrended snow depth residuals in the Hourglass (HG) and meadow (MD) sites with colors representing different resolutions. Black dots represent median value and color shades represent the distribution of points for each resolution on each sampling day.

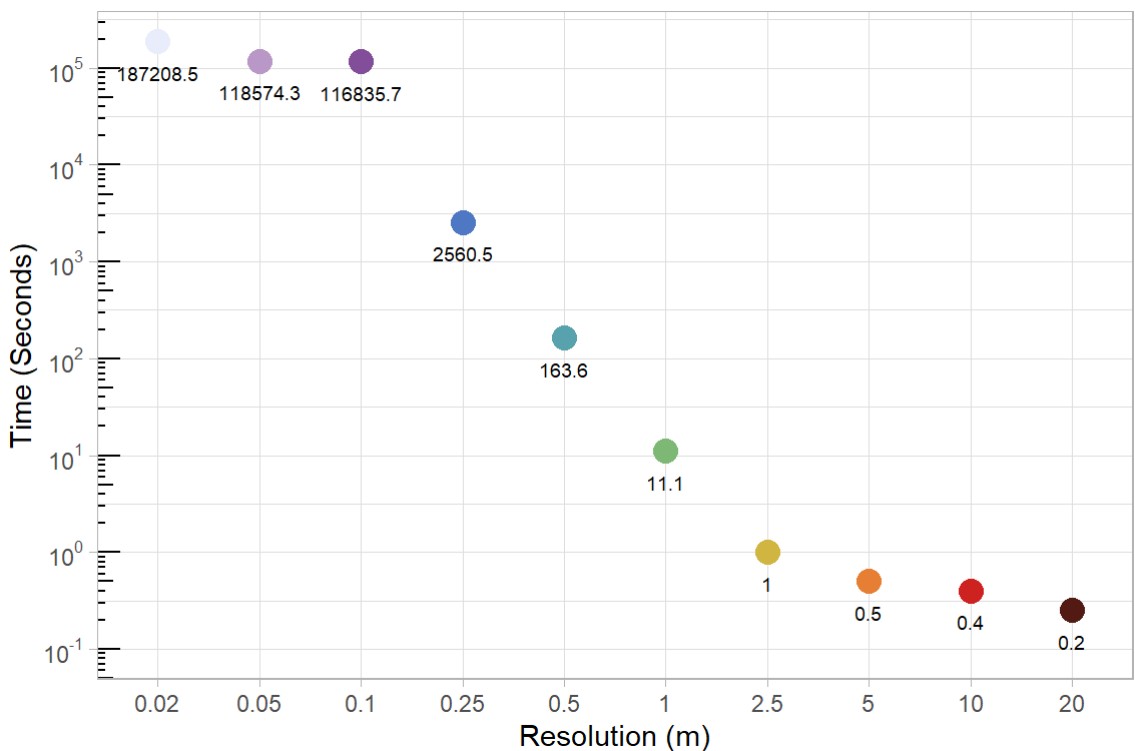

Figure A10: Variogram processing times for the 27 February 2020, observation day. Colors indicate different snow depth digital surface model (DSM) resolutions and the number labels are the variogram processing times (in seconds) for each DSM. Note that fully processing the 0.05 and 0.02 m grids was too computationally expensive even when using a supercomputer. Therefore, we randomly sampled three million points from those DSMs before calculating the variograms. Thus, the processing

 times for 0.05 and 0.02 m are similar.

**Code and Data Availability**

The timeseries of snow depth digital surface models (before vegetation masking, detrending, and outlier removal), the vegetation masked Hourglass and meadow shapefiles and .csvs of data for the figures presented are available in a US Geological Survey data release, located at: https://doi.org/10.5066/P9YCIA1R.

**Author Contributions**

ZM, EP, and ES designed the study. ZM and RP collected the dataset. ZM, EP, and KB drove the theoretical discussion. ZM prepared the manuscript with contributions from all co-authors.

**Competing Interests**

The contact author has declared that neither they nor their co-authors have any competing interests.

**Disclaimer**

Any use of trade, firm, or product names is for descriptive purposes only and does not imply endorsement by the U.S. Government.

**Acknowledgements**

The authors would like to acknowledge the support of the Custer-Gallatin National Forest US Forest Service office and Bridger Bowl Ski Area for permitting the research be conducted on their managed lands. The authors would like to acknowledge a review of an earlier version of this work by Jeffrey Deems and the fieldwork assistance of Madeline Beck, Gabrielle Antonioli, Zachary Keskinen, Jordy Hendrix, and Grete Gansauer.

**Financial Support**

This research was supported by the U.S. Geological Survey Land Change Science/Climate R&D Program, Montana State University's Earth Science Department, and the Montana Association of Geographic Information Professionals (2020 scholarship).

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
