# Peer review of "Assessing the Seasonal Evolution of Snow Depth Spatial Variability and Scaling in Complex Mountain Terrain"

_The Cryosphere, 2022_

## Author Comment (AC1)

**Response to Referee Comments**

**Submission Title:** *Assessing the Seasonal Evolution of Snow Depth Spatial Variability and Scaling in Complex Mountain Terrain*

**Submission Number:** *tc-2022-96*

Thank you for your time providing a thorough review of this manuscript. We appreciate your feedback on the manuscript and the relevance you find in the work. We addressed each comment and provided an updated version of the manuscript as well as responses to each comment below.

**Reviewer 1 (Yves Bühler) Main Points:**
1. The applied nearest neighbor resampling technique is in my opinion the wrong approach to resample the different snow depth maps. If nearest neighbor is taken, the value of the coarser resolution grid is the value that is located closest to the center of the new cell. If you go to coarser resolutions (0.5 – 20 m) this makes no sense as this value can be very random. I would propose an aggregation or a cubic convolution resampling.

> We chose the nearest neighbor resampling technique to avoid over-smoothing the dataset and to maintain the naturally occurring variability found in spatially continuous snow depth. Our initial results used an aggregation method for resampling, which we found to over-smooth the inherent variability in snow depth, especially at < 1 m scales, resulting in unrealistically smooth experimental variograms with less defined ranges. Additionally, we did not want to add any additional abstraction, and therefore uncertainty, to the detrended snow depth values used in our analysis. Cubic convolution techniques may result in cell values outside the range of the input raster, especially near edges, which our vegetation masked dataset has many of and we did not want to introduce this uncertainty into our spatial variability analysis. We included additional clarification and supporting citations to the manuscript on lines 234-239.

2. There are no figures illustrating the snow depth maps or the applied detrending. It would be important for the readers to see such figures here to better understand what is done.

> We added a figure clarifying the vegetation masking and detrending steps (Figure 3) as well as figures of the timeseries of detrended snow depth maps at the two locations to the manuscript appendix (Figures A3 and A4).

3. The motivation, why semivariograms are used, is not really clear. Are there other possible methods? If yes, a comparison of the results from other methods would be very interesting.

> We set out to try to assess the appropriate scale for measuring snow depth in complex terrain. We were motivated to observe the spatial structure at different resolutions that allows us to reliably interpolate between data points while capturing the naturally occurring variability of snow depth. Variograms are the generally accepted tool for assessing the scales of variability with spatial data and have been widely used in the snow depth spatial variability literature. Therefore, we chose this method because it is well understood and makes our results comparable to previous work. We added this justification and relevant citations to the manuscript on lines 219-233.

4. The discussion is very much based on hard-to-understand metrics (Sill, Range etc.). I am missing a part where the discussion is on a level where the average reader can follow. What do these values mean discussed on examples, best illustrated with figures showing the snow depth distribution.

       We added definitions of these values into the methods section (Sec. 3.5, lines 224-231).

5. The investigated site is very small and we do not know how representative this is. It is not clear if the findings that are presented are valid for further regions. We would have drone-based snow depth maps from sites in the region of Davos, Switzerland (also several dates in one winter) we could provide to check if the results are consistent in different regions.

       We agree that complex terrain is challenging to effectively prove as broadly representative and have clarified our language throughout the paper to specify the complex terrain within our study site (lines 20, 89, 396, 407, 428, 456, 469, 472). It would be very relevant to see if our results are similar to those at other sites (such as Davos) and we would be interested in pursuing such collaborative research in the future. Addressing necessary computational processing power would be a key component of adding additional sites in future research, given how computationally expensive high-resolution large scale variogram processing is.

---

## Author Comment (AC2)

**Response to Referee Comments**

**Submission Title:** *Assessing the Seasonal Evolution of Snow Depth Spatial Variability and Scaling in Complex Mountain Terrain*

**Submission Number:** *tc-2022-96*

Thank you for your time providing a thorough review of this manuscript. We appreciate your feedback on the manuscript and the relevance you find in the work. We addressed each comment and provided an updated version of the manuscript as well as responses to each comment below.

**RC2 (Anonymous) Major Points:**
1. It is not clear why spherical models are used. There are other models which can fit omnidirectional variograms. Please check references, test other models (log linear models, exponential…) and justify why you use one particular model. Similarly I think the experimental omnidirectional variograms must be shown in the manuscript to justify the fit with different models.

> We chose to use spherical models because they are well suited for three-dimensional spatial analysis, are commonly used in similar variogram analysis (making our work more comparable to previous literature) and fit the majority of our 170 experimental variograms, especially in the complex terrain of the Hourglass. Given that we calculated and compared 170 different variograms, we needed to pick one model to compare each of them with. Therefore, the spherical fit was the right choice overall despite the possibility of better fits with different models for some of the variograms. We added clarification and citations to the manuscript (lines 241-243). We also included a figure of the timeseries of experimental variograms to the appendix for reference (Figure A2).

2. Methods section needs further details on how variograms are computed (references, mathematical expressions). Moreover, there are some methodological issues that must be corrected in the analysis. For instance, the maximum distance considered on variogram computation cannot be larger than half the maximum points pair distance (Sun et al., 2006 https://doi.org/10.1080/01431160600676695). Methods section needs a full description of the variables later analyzed (range, nugget, sill) in view to the vocabulary of previous works. Finally why scale breaks are not computed?).

> We added a formula and description of how our variograms are computed as well as the definitions of the key components of a variogram (sill, range and nugget) to the manuscript (lines 219 – 231). The maximum distance in our experimental variogram calculations was 1/3 the length of the diagonal of the box spanning the data (R "gstat" package - https://cran.r-project.org/package=gstat), rather than the maximum distance of point pairs as described previously (line 244). We clarified this in Sec. 3.5. Scale breaks were considered in initial questioning but would require additional fractal analysis (outside the scope of this paper) and were not calculated. The "scale breaks" verbiage was changed to "spatial scales of snow depth variability" in the only location it was mentioned in reference to our work (line 16) but is still included in reference to other works which did calculate and define the scale breaks.

3. I encourage manuscript authors to show the results for all resolutions and not only for 0.5 m. Please include values of table 2 for all resolutions. Also include results of figure 7 for all resolutions.

> Table 2 has been updated with mean values for each resolution and location. Figure 8 (formerly Figure 7) has been updated to include all resolutions.

4. From the results shown in the manuscript, I consider it is not justified this conclusion: "We found that spatial resolutions greater than 0.5 m do not capture the complete patterns of snow depth spatial variability within complex mountain terrain". For some dates this is true but in most of them 1 m resolution results (figure 5) is quite close to 0.5, 0.025, 0.1m…, and in some cases also 2.5m. This point is no convincing from my understanding. Additionally, a steep couloir is a very characteristic "complex mountain terrain", so I think it is not possible to extend this conclusion to "all complex mountain terrain". Please change conclusions conveniently.

> Indeed, most dates' snow depth spatial variability patterns are captured by 1 m resolution sampling strategies quite well, but we clearly see that it doesn't capture all instances of spatial variability at our site. Therefore, we recommend the coarsest resolution which captures all finer resolution patterns of spatial variability, 0.5 m, as a minimum sampling resolution. We agree that complex terrain is challenging to effectively prove as broadly representative and have clarified our language throughout the paper to specify the complex terrain within our study site (lines 20, 89, 396, 407, 428, 456, 469, 472).

**RC2 Minor Points:**
Line 15: "We produced 12 snow depth maps…" I think here and all long the manuscript there is an error. Figures show 11 snow depth maps. Maybe authors refer here to 12 UAS flights (including the snow-free flight), from which 11 snow depth maps were derived. Correct conveniently in the manuscript (introduction, discussion, conclusion…

> Thank you for catching this typo. Revised to 11 snow depth maps.

Line 50-55: I consider that, here, it is needed to state slope limitations of satellite-derived snow maps, which cannot be obtained when the slope is above a threshold value. Please verify this threshold in the references and include it.

> We added "Yet the accuracy of DEMs produced through satellite imagery and stereoscopic processing are known to suffer on slopes steeper than 35°, common in high-relief mountain terrain (Lacroix, 2016; Shean et al., 2016, Deschamps-Berger et al., 2020)." (lines 53-55).

Line 81: Change "sparse temporal observations" to "more sparse temporal observations" and include other works analyzing the snow depth spatial variability from UAS/LiDAR systems. Here the links to some examples: https://doi.org/10.1029/2020WR027343, https://doi.org/10.1016/j.jhydrol.2019.124046

> We added "more" to the sentence and additional citations (line 84).

Line 113-119: Three AWS are described here, however in Figure 2 only data from Brackett meadow AWS are included. Why you describe the other two? Remove the description of AWS not used in your research please.

Updated to only include one AWS (lines 118-121). We originally described the other two AWS since they are used to report Bridger Bowl Ski Area's average annual snowfall and were useful for observing hourly meteorological data during data collection.

Line 132: The grid pattern at 50 m above ground was respected for the entire flight or this elevation was programmed in the UAS control software in view to the elevation data of this later software? Please specify.

The grid pattern at 50 m AGL constant height was pre-programmed into the UAS flight control software to follow terrain from an input 1 m resolution DSM captured prior to winter flights. We have updated the text with additional clarification (line 135).

Line 133: change units of ground sampling distance to the cm/pixel as this is the most extended unit used in UAS works.

Updated to 2 cm (line 136). We chose to use only meters for consistency throughout the paper but agree that ground sampling distance is most easily compared in cm/pixel.

Line 139: How many points of the 25 ground control points were included in the partial selection? Were used, 1 to 3, 5 to 25, at least 7? This information is relevant for further error assessment.

We typically ended up using between 3 and 10 points for each model and updated the text to include this (line 142).

Line 140: Similarly to previous point, how many surveyed snow depth points were deployed each date? Please include this information.

We collected manual snow depth measurements at 4 or more locations at deployed 1 m$^2$ markers each field day (lines 144 and 176).

Line 153: Include the horizontal and vertical RMSE of image geolocation after PPK processing. Later, include details on the RMSE values obtained for the GCPs and the number of GCPs available for validation each date.

We have added a table of after-PPK processed image location differences (Table A1).

Line 163: Also Include references to other works evaluating RTK and GCP accuracies in snow dominated areas: Eberhard et al., 2021 (https://doi.org/10.5194/tc-15-69-2021) and Revuelto et al., 2021 (https://doi.org/10.1029/2020WR028980)

We previously mentioned Revuelto et al., 2021 in regards to RTK and GCP accuracies but have added the Eberhard et al., 2021 citation to the sentence as well (lines 165-166).

Line 164: Did you obtain the snow depth distribution directly subtracting regular grids from snow free and snow covered DSMs?, Were these differences computed with 3D point clouds and then rasterized to the distinct spatial resolutions? Please detail this. If 3D point clouds were directly subtracted detail the software used.

As stated in the methods (lines 167-169), we simply used regular aligned grid subtraction of the original snow-free from snow-covered 2 cm resolution DSMs to find our snow depths. The DSMs were derived from Agisoft Metashape processing and the DSM differencing was completed in R. We did not use 3D point clouds for differencing.

Line 166: Please give an idea of what unrealistic snow depths are for you. I guess these are negative values, too high values…

> We added clarification "Unrealistic snow depths are negative snow depth values and values filtered by expert judgement" (line 172).

Line 174-177: I think that the manual avalanche crown profile data are not used in this research so I do not see any need to include this information. I would remove it.

> We use the avalanche crown profile as a point snow depth measurement and it is the single validation measurement we have from the upper reaches of the Hourglass during the season (lines 287-290).

Line 195: I think manuscript authors are working with regular grids, but this is not clearly stated in the manuscript. I consider it is needed to clearly state this in the first sentence of the DSM detrending section.

> We added clarification "…we detrended each snow depth DSM with regular grids to focus analysis on the residual surfaces…" (line 201).

Section 3.5 Variogram calculation must be detailed as this is one the main analysis of this work. Include suitable references here and the mathematical expressions to compute them. I think it is unbalanced the details given in section 3.6 for computing the coefficient of variation whereas no details are given for variogram calculation.

> We added a formula and description of how our variograms are computed as well as the definitions of the key components of a variogram (sill, range and nugget) to the manuscript (lines 222-233).

Line 213: Why spherical models? there are other models that can be tested and fitted to the variogram. See Mendoza et al., 2020 (https://doi.org/10.1029/2020WR027343). Indeed there are other models which fit better the snow behavior depending the spatial scale (see Noriaki et al., 2019 https://doi.org/10.1002/hyp.13415, Mendoza et al., 2020 https://doi.org/10.1029/2020WR028480) . This point (also referred in major points section) must be fully addressed and justified.

> We chose to use spherical models because they are well suited for three-dimensional spatial analysis, are commonly used in similar variogram analysis (making our work more comparable to previous literature) and fit the majority of our 170 experimental variograms, especially in the complex terrain of the Hourglass. Given that we calculated and compared 170 different variograms, we needed to pick one model to compare each of them with. Therefore, the spherical fit was the right choice overall despite the possibility of better fits with different models for some of the variograms. We included additional clarification and citations to the manuscript (lines 241-243).

Line 218:  It is not possible to compute variograms till the maximum distance of the study area, it must be half of the maximum point pairs distance (Sun et al., 2006 https://doi.org/10.1080/01431160600676695). This is an important point that must be also addressed in addressed in results and methods section.

The maximum distance in our experimental variogram calculations was 1/3 the length of the diagonal of the box spanning the data (R "gstat" package - https://cran.r-project.org/package=gstat), rather than the maximum distance of point pairs as described previously. We clarified this in Sec. 3.5. and updated the manuscript (line 244).

Line 223: Include details on how the "average" variogram is computed. Later, in results section (figure 4). The seasonally averaged variogram is shown. Nonetheless no details on how is computed are included (each point is the average of semivariances for all resolutions?)

Updated text with the addition of: "We applied a local polynomial regression (LOESS) fit to spherical models to produce the seasonally averaged resolution specific variograms (R Core Team, 2021)" (lines 248-250).

Line 235: The n=70 snow depth measurements include all observations? This is, following Lopez-Moreno (et al., 2011), 70/(4 corners+center obs) = 14 locations? Or the 70 snow depth observations repeated this procedure in 70 distinct locations? Please clarify. Where these locations constant all acquisition dates?, always the same number?

70 manual snow depths measurements were collected throughout the entire season with 5 individually probed snow depths collected at each marker (4 corners + 1 center = 350 total individual measurements). The 70 snow depth observations were each randomly distributed throughout the avalanche-safe lower areas of the study site. We deployed and measured at least 4 snow depth validation markers each field day in random locations. The manuscript has been updated to clarify this (lines 174-179, 286-287)

Also an overview of the locations of these validation locations would help to potential readers where did you validate your UAV observations. I encourage manuscript authors to include one snow depth distribution map (of one selected date, maybe the seasonal maximum) to allow the reader to see the snow distribution characteristics. Include in this new figure the point snow depth measurement locations.

We added a complete timeseries of detrended snow depth maps of the Hourglass (Figure A3) and meadow (Figure A4) that clearly demonstrate the distribution of snow depths throughout the research sites. We state that "We collected all validation snow depth measurements, except the crown profile of the 26 February 2020, avalanche, in the lower elevations of the Hourglass and throughout the meadow in order to avoid exposure to snow avalanches" and clarify that "This assessment is not a comprehensive assessment of error because our validation snow depths were primarily collected at random locations in the safe lower slopes, and this assessment is therefore biased towards comparisons of measurements in the meadow" (lines 283-290).

Line 268: I think the experimental variogram must be included in this comparison. This is, show both, the spherical fit variogram and the variogram derived from true snow depth observations. This might demonstrate that spherical model fits the experimental variogram. This is related with the major point of models tested.

We added a figure of the timeseries of experimental variograms to the appendix for reference (Figure A2).

Line 269: Please define in methods section what is for you range (I guess scale breaks but not sure), nugget and still. These details can be included in section 3.5. Why scale breaks are not computed?

> We added the definitions of the key components of a variogram (sill, range and nugget) to the manuscript (lines 222-233). Scale breaks were considered in initial questioning but would require additional fractal analysis (outside the scope of this paper) and were not calculated. The "scale breaks" verbiage was changed to "spatial scales of snow depth variability" in the only location it was mentioned in reference to our work (line 16) but is still included in reference to other works which did calculate and define the scale breaks.

Line 272-273: If 0.5 m resolution is representative of most resolutions, you must somehow show it. Table 2 admits more information, so please include range, nugget and semivariance (average semivariance??) for all resolutions in both sites. Otherwise you cannot say that 0.5 m resolution is representative. If manuscript authors consider that this information is not needed in table 2, this information must be included in the supplementary material.

> Table 2 has been updated with mean values for each resolution and location.

Line 275: I think all the information of Figure 3 is not fully described in just one sentence. For example the variogram models differences in HG, MD for the 15/01/2020 deserve some comments. Similarly why this marked difference in 20 m resolution the 17-03-2020? Provide more detailed comments on this figure.

> Considering Figure 4 (formerly Figure 3), we added "These results reflect the given substratum of the two sites. The meadow's more homogenous ground cover and topography are reflected in less variability overall and spatial autocorrelation over greater distances. In contrast, the steep rocky terrain of the Hourglass is reflected in the more dynamic seasonal patterns of spatial variability and shorter distances of autocorrelation. The 20 m resolution variograms frequently misrepresent the spatial variability patterns of finer resolutions and this is perhaps due to the relatively small study sites creating far fewer point pairs of snow depths to calculate the variograms from, therefore being less resistant to outliers." (Sec. 4.2, lines 310-315).

Line 289: "the greatest variability" in what? Between the scale breaks, between the resolutions?

> Updated to read: "the greatest semivariance values exist earlier in the winter at both the Hourglass and meadow at all resolutions" (line 331).

Line 308: In HG, I would say that for 10 out of the 11 snow depth observations, 1m resolution has a range close to that of 0.5, 0.25 m. In the worst case, only for one date (the third acquisition on February) the range value is close to that of 1m, 2.5 m…This is not a difference of 25-75 % of the observation dates. This result does not fully support one of the major outcomes of this research: 1 m is not enough to capture snow depth spatial distribution. This happens in some cases from my understanding of your results.

> The 1 m resolution range values on 21 Feb and both May observations days is quite different than 0.5 m resolution range values (Figure 6) which results in different ranges on 27% of days observed in the Hourglass (3 of 11). We updated the text to clarify the specific instances where 1 m resolution does not represent the higher resolution patterns (lines 349-352). Snow depth spatial variability patterns on most days are captured by 1 m

resolution sampling strategies quite well, but we clearly see that it doesn't capture all instances of spatial variability at our site. Therefore, we recommend the coarsest resolution which captures all finer resolution patterns of spatial variability, 0.5 m, as a minimum sampling resolution.

Line 321: Show in a new figure or table CV coefficients for all resolutions. This might justify that the "coefficients of variations were similar across a variety of snow depth DSM resolutions" and also will justify why you choose 0.5 m resolution choice.
> We updated Figure 8 (formerly Figure 7) with CV results from all resolutions.

Line 346-350: You cannot argue that your results suggest that elevation is the most significant variable and justify that the natural elevation gradient runs parallel to the dominant wind direction. I encourage manuscript authors to compute directional variograms for 8 distinct angular windows. In the contrary I consider this point cannot be fully discussed here.
> We revised the sentence to more clearly reflect discussion on finding elevation as the most significant predictor for directional bias in our dataset and removed mention of wind direction from our results (lines 391-392).

Line 355: In view to results of figure 3, 4 and 5, I do not agree with this statement. Some days 0.5 captures well the spatial variability, yes, but for many days also 1m and even 2.5 m.
> Snow depth spatial variability patterns on most days are captured by 1 m resolution sampling strategies quite well, but we clearly see that it doesn't capture all instances of spatial variability at our site. Therefore, we recommend the coarsest resolution which captures all finer resolution patterns of spatial variability, 0.5 m, as a minimum sampling resolution. We have revised the sentences to clarify our findings and discussion (lines 396-340).

Line 359: I think that conclusions about range values between the different resolution cannot be based on the average calculations (figure 4 and 6). These are important but, it must be underlined that for many dates (see figure 5) range and sill or resolutions from 0.02 to 1 and sometimes 2.5m are quite close.
> We updated the text to clarify that 1 and 2.5 m resolution models have similar ranges on some observation days and provided further clarification of average range values (Table 2 and lines 402-405).

Line 362: In view to previous comments, I do not agree with the statement about 0.5 m resolution. This resolution is needed to capture small-scale spatial variability for certain dates, but in most of them, 1 m (and for some of them 2.5 m) is enough to capture the spatial variability in complex terrain (HG).
> We updated lines 396-400 and 405-407 to clarify these findings are for our study site and that 0.5 m resolution is the coarsest resolution required to capture all small-scale variability. Most dates' snow depth spatial variability patterns are captured by 1 m resolution sampling strategies quite well (7 of 11), but we clearly see that it doesn't capture all instances of spatial variability at our site. Therefore, we recommend the coarsest resolution which captures all finer resolution patterns of spatial variability, 0.5 m, as a minimum sampling resolution.

Also I think you cannot extend your results to "complex terrain" you are covering a domain with very particular characteristics, a steep couloir.

> We agree that complex terrain is challenging to effectively prove as broadly representative and have clarified our language throughout the paper to specify the complex terrain within our study site (lines 20, 89, 396, 407, 428, 456, 469, 472).

Line 372: Here you talk about scale breaks, but it is not clear if you compute these or not. Please see my comment on line 269.

> Scale breaks were considered in initial questioning but would require additional fractal analysis (outside the scope of this paper) and were not calculated. The "scale breaks" verbiage was changed to "spatial scales of snow depth variability" in the only location it was mentioned (line 16).

Line 374: Which is the value of the similar consistency in less than 20 m? Can you please give the scale breaks you obtain?

> We consistently found range values of the Hourglass to be < 20 m throughout the winter (seen in mean range values in Table 2). We did not calculate scale breaks and have updated the manuscript to clarify this (lines 419-420).

Line 403 to 414: I consider this paragraph is not really needed. This is computation, not snow since. Please reduce this paragraph and give a (very) short overview of this point.

> We shortened the paragraph to simplify the message (lines 449-454), but still believe that the exceptional amount of processing time required is relevant to discuss in light of future work.

Conclusions: Change conveniently in view to previous comments. These conclusions must be changed accordingly to the spatial resolution needed for accurately capturing snow spatial variability, which, regarding your results are in most cases 1 m and for some dates 0.5m. Moreover the conclusions must highlight that this work was conducted in a steep couloir, which is a frequent landform in complex mountain terrain but not the only landform.

> We appreciate your interpretation and discussion of these findings. We have clarified throughout the results and discussion that 1 m resolution captures the patterns on most days at our site but does not for some dates (as you have stated above). Therefore, we still conclude that 0.5 m resolution sample spacing is required to capture the snow depth spatial variability within our site every single time. We updated lines 464-465 to further emphasize future research in additional mountain terrain and snow climates.

Figure 1: Include a photograph of the Meadow area to allow readers having a clear idea of differences between Hourglass couloir and the meadow area. I also encourage manuscript authors to include an intermediate map between the lower right corner USA frontiers and the topographic map. Please change figure composition and include a colored DEM of the mountain range where the location of the study site can be better observed.

> We updated Figure 1 to include an overview of the Bridger Mountain Range and a photograph of the meadow research area.

Figure 3: This figure is difficult to understand. Please split it in two more rows and increase subplots size for an easier understanding.

> Thank you for your suggestion on modifying Figure 4 (formerly Figure 3). We considered this suggestion at great length and modified it slightly. However, adding more rows increases the complexity of the figure. We believe the current emphasis of presenting variance over time (11 column time series) and space (2 main sites represented in the rows) is the best means of presenting these data and results. Adding more rows would create confusion and not allow the reader to view the variance differences between the two sites through time as easily. However, we modified the figure to include darker plot boundaries for each day's plot so that the x-axis values (Distance (m)) are more clearly delineated.

Figure 6: Include a legend for the colors of the points.

> Although the colors were representative of the x axis resolutions because they match the color legends (color = resolution) of all other figures in the manuscript, the colors are not essential in the interpretation of Figure 7 (formerly Figure 6) and we have removed color from the plot.

---

## Author Response (AR3)

**Response to Referee comments**

**Submission Title:** *Assessing the Seasonal Evolution of Snow Depth Spatial Variability and Scaling in Complex Mountain Terrain*

**Submission Number:** *tc-2022-96*

Thank you again for your time providing an additional review of this manuscript. We appreciate your feedback on the manuscript and the relevance you find in the work. We addressed each comment and provided an updated version of the manuscript as well as responses to each comment below.

**Reviewer 1 (Yves Bühler) Suggestions:**

Dear Authors

Most of my points are satisfactorily answered by your revision. However the most critical methodological point is not.

You argue that you use the nearest neighbor resampling to "not over-smooth" the snow depth values. But resampling from high (2 cm) to low (0.5 - 20 m) with nearest neighbor is simply wrong and has assumingly a very large impact on the results and conclusions of your study.

To compare remotely sensed snow depth values at different spatial resolution you need to aggregate the HS values within the area of the larger resolution cell. Applying nearest neighbor resampling just takes the exact value of the high spatial resolution cell that is closest center point of the lower spatial resolution cell. And this is clearly wrong.

The real snow depth within a lower resolution cell is very different from the center value in particular within complex terrain (which you investigate). The deviations from the real snow depth can easily reach more than 100 %. The center value is very random if your snow depth distribution is not perfectly homogenous.

Due to this methodological flaw which probably has a very large impact on the results unfortunately, I cannot recommend the paper for publication.

**Response to Reviewer 1 Suggestions:**

We appreciate your perspectives and comments regarding resampling methods and completed further analysis of the effects of resampling method on spatial variability results. In addition to using the nearest neighbor (NN) method in our original analysis, we resampled our 2cm resolution DSMs to each resolution using the cubic convolution (CC) technique and completed variogram calculations on the resultant DSMs. We found that similar patterns for spatial variability exist in the experimental variograms using both NN and CC resampling methods. However, the total semivariance is consistently lower in the complex terrain of the Hourglass (Fig. A3). These subtle differences in the experimental variograms have less subtle effects on the spherically fit models which, in some cases, miss the initial short-range-distance sill (15 ~ 20 m) and instead fit to a much larger range (~ 60 m) with the accompanying increase in sill value to match (Fig. A4, A5, and A6). This reinforces our idea that the cubic convolution resampling methods over-smooths and decreases the naturally occurring variability observed in the dataset of the

complex heterogenous terrain of the Hourglass. Alternatively, results are nearly identical in the homogenous terrain of the meadow when using NN and CC techniques.

To further assess whether the differences in variability using different resampling techniques are attributed to over-smoothing using CC or artificially inflated using NN (as suggested by Reviewer 1) we resampled the DSMs using NN, CC, and mean and median aggregation methods. We then completed a pairwise correlation analysis of paired points of each method to observe any systematic bias in the resampling method. We found that all of the resampling methods result in highly correlated ($> 0.97$) point pairs for all spatial resolutions $< 1$ m. Correlation values decrease markedly as the resampled resolution increases beyond 1m between nearest neighbor and all other methods decreases (Fig. A2). Therefore, resampling methods should be considered closely when resampling to resolutions greater than 1m. Additionally, while the nearest neighbor method utilizes only observed values, the cubic convolution method calculates derived values and our results suggest this method produced unreal values in highly variable areas (ex. along an avalanche crown on Feb 27).

Our original results suggest spatial resolutions $< 0.5$m are necessary to observe the complete picture of spatial variability at our site. Given the high correlation of paired values across all resampling methods at resolutions $< 1$ m, we show that the nearest neighbor technique does not artificially inflate the variance and captures the naturally occurring spatial variability of snow depth within the high-resolution context of our study. Overall, through our original analysis and this additional analysis, we are able to identify a resolution (1m) at which results start to diverge based on resampling technique, identify a resolution (50cm) at which spatial variability can be captured independent of resampling technique in the complex (and homogenous) terrain of our study site, and, finally, identify the distance at which snow depth differs across the complex mountain terrain of our study site(15 m).

We've included additional text, citations and figures regarding the resampling re-analysis in these locations of the manuscript:

- Sec 3.5: Variogram Calculation and Fit - lines 239 – 246
- Sec 4.2: Resampling Results – lines 313 – 323 (new section)
- Sec 4.3: Variogram Results – lines 326 – 331
- Sec 5.2: DSM Resampling Methods – lines 424 – 445 (new section)
- Sec 6: Conclusions – lines 524 – 525
- Fig A2 – A6 (additional figures comparing resampling results)

**Reviewer 2 (Anonymous) Suggestions:**

Review of the manuscript "Assessing the Seasonal Evolution of Snow Depth Spatial Variability and Scaling in Complex Mountain Terrain"

I must congratulate manuscript authors in view to the changes included in the manuscript after the major review. They have conveniently discussed all points raised by reviewers, changing the manuscript where appropriate or justifying their choices. However I have some minor points that I would like to point out:

1. Figure 3 maps: I encourage manuscript authors to include classified colors scales. The colors for snow depth and elevation can be used, but continuous color scales are difficult to interpret. 7 to 9 snow depth classes can help to more easily see the snow depth variability. For instance Figure A3 scale bar has more resolution than that of Figure 3. If possible make thicker (or darker) contour lines in Figure 3.

2. I agree that, spherical models, are commonly used in variogram analysis based on the references you provide. However there are other models with great potential. You might cite other models used in snow science with good result (log linear, exponential...) and state that you have finally used spherical. In this regard I think the comparison between experimental variograms and fitted variograms is needed. Indeed you state you have compared them (Line 304-305), but no adjustment metrics are shown (R2 or others....) or at least plot them together (in Figure A2 include right below experimental variograms the fitted ones). Please include one of these previous suggestions.

3. Manuscript authors have chosen a 1/3 length of the diagonal box and they support their decision on R "gstat" package URL documentation. This is a bit difficult to find in this documentation. Can you please help potential readers to find this with other references? This 1/3 choice deserves some discussion since, as far as I know, in snow science 1/2 distance is usually applied. Just discuss if some differences could be found if a different maximum distance is applied to semivariogram computation in view to previous works.

4. Figure 8: Circles superposition makes difficult the interpretation, Might you reduce circles size and allow some transparency?

5. Conclusions: I think it is worth to change a bit conclusion section, stating (line 471): "Despite for more than one half of our UAV acquisitions 1m sample spacing is able to capture the natural snow spatial variability, to guarantee the full capture of snow depth variability for all observation dates, 0.5 m sample spacing is required….." or a rephrase of this sentence

6. In previous figure 6 comment, there was a misunderstanding. Previous Figure 6 (now figure 7) was fine. I meant that a legend with circles color correspondence can help to interpret it (as included in figure 6 of final manuscript version).

**Response to Reviewer 2 Suggestions:**

**Thank you for a second thorough review, and we appreciate your sentiments. We addressed your comments as follows:**

1.  Figure 3 maps have been updated to include discrete classified color scales and clearer elevation contours.
2.  An additional comment and citation have been added regarding other variogram model fits (lines 236 – 237) and a table of RMSE and NMAD values for the spherical fits has been included in the appendix (Table A5). Comparative model fitting lies beyond the scope of the paper largely due to the additional extensive computational requirements of additional fits.

3. We have added additional discussion of these choices and citations of other works (lines 250 – 255).
4. Figure 8 has been updated with smaller and transparent points jittered around the shaded collection dates for improved interpretation.
5. We state this distinction in the results and discussion and have left it out of the Conclusion for brevity and readability.
6. Figure 7 has been updated with colored points and legend.